# Molecular Manipulation of the miR396 and miR399 Expression Modules Alters the Response of *Arabidopsis thaliana* to Phosphate Stress

**DOI:** 10.3390/plants10122570

**Published:** 2021-11-24

**Authors:** Joseph L. Pegler, Duc Quan Nguyen, Jackson M. J. Oultram, Christopher P. L. Grof, Andrew L. Eamens

**Affiliations:** 1Centre for Plant Science, School of Environmental and Life Sciences, Faculty of Science, University of Newcastle, Callaghan, NSW 2308, Australia; joseph.pegler@newcastle.edu.au (J.L.P.); ducquan.nguyen@uon.edu.au (D.Q.N.); jackson.oultram@uon.edu.au (J.M.J.O.); Chris.Grof@newcastle.edu.au (C.P.L.G.); 2Institute of Genome Research, Vietnam Academy of Research and Technology, 18 Hoang Quoc Viet Str., Cau Giay, Hanoi 100000, Vietnam; 3School of Science, Technology and Engineering, University of the Sunshine Coast, Maroochydore, QLD 4558, Australia; 4School of Chemistry and Molecular Biosciences, The University of Queensland, Brisbane, QLD 4072, Australia

**Keywords:** *Arabidopsis thaliana* (*Arabidopsis*), phosphorous (P), phosphate (PO_4_) starvation, molecular manipulation, microRNA396 (miR396), miR399, RT-qPCR gene expression analysis

## Abstract

In plant cells, the molecular and metabolic processes of nucleic acid synthesis, phospholipid production, coenzyme activation and the generation of the vast amount of chemical energy required to drive these processes relies on an adequate supply of the essential macronutrient, phosphorous (P). The requirement of an appropriate level of P in plant cells is evidenced by the intricately linked molecular mechanisms of P sensing, signaling and transport. One such mechanism is the posttranscriptional regulation of the P response pathway by the highly conserved plant microRNA (miRNA), miR399. In addition to miR399, numerous other plant miRNAs are also required to respond to environmental stress, including miR396. Here, we exposed *Arabidopsis thaliana* (*Arabidopsis*) transformant lines which harbor molecular modifications to the miR396 and miR399 expression modules to phosphate (PO_4_) starvation. We show that molecular alteration of either miR396 or miR399 abundance afforded the *Arabidopsis* transformant lines different degrees of tolerance to PO_4_ starvation. Furthermore, RT-qPCR assessment of PO_4_-starved miR396 and miR399 transformants revealed that the tolerance displayed by these plant lines to this form of abiotic stress most likely stemmed from the altered expression of the target genes of these two miRNAs. Therefore, this study forms an early step towards the future development of molecularly modified plant lines which possess a degree of tolerance to growth in a PO_4_ deficient environment.

## 1. Introduction

In plant cells, the molecular and metabolic processes of nucleic acid synthesis, phospholipid production, coenzyme activation, and the generation of considerable amounts of chemical energy in the form of ATP (adenosine triphosphate) and GTP (guanosine triphosphate), all rely on an adequate supply of the essential macronutrient, phosphorous (P) [1,2]. The requirement of an adequate supply of P in plant cells is readily evidenced by the intricately linked molecular mechanisms of P sensing, signaling and transport throughout plants [3,4], together with the severely impeded developmental progression of plants when cultivated in P limited or deplete conditions, including altered root architecture, inhibited shoot elongation, and the over-accumulation of the antioxidant pigment, anthocyanin [5]. Phosphate (PO_4_), in the form of inorganic phosphate (Pi), is the predominant form of P taken up from the soil by the plant root system; however, soil PO_4_ primarily exists in an organic or insoluble form which is largely inaccessible by the uptake mechanisms of plant roots [6]. Due to the limitation of available soil PO_4_, combined with the absolute requirement of an adequate concentration of P in all plant cells for normal growth and development, plants employ elegant mechanisms to spatially regulate the cellular concentration of P across their developmentally distinct tissues [7,8]. To achieve such spatial variance in cellular P concentration, plants continually modulate P homeostasis via constant regulation of the rate of acquisition of external PO_4_ from the soil, in combination with the parallel adjustment of the degree of remobilization of their existing internal stores of P [7,9].

It is predicted that up to 40% of the yield of the world’s most economically significant crop species is limited by poor soil P availability [10]. In modern agriculture, this is addressed via the application of P fertilizers, a practice which itself presents a serious environmental concern with such fertilizers generated from finite and rapidly depleting natural deposits of phosphorite, deposits predicted to be completely exhausted within the next 50 to 100 years [11,12,13]. Further compounding this issue is that the major crop species only take up 15 to 30% of the exogenously applied P within the first 12 months of application, with the remainder running off or leaching into the local terrestrial, and subsequently aquatic ecosystems, which can lead to the eutrophication of habitats proximal to agricultural land [14,15]. Of additional concern is the continual and ever more rapid expansion of the global population, population growth that is predicted to require approximately double the current crop production outputs by 2050 to meet consumer demands on agriculture [16,17,18]. One avenue being pursued to attempt to adequately address this growing demand is the use of a molecular engineering approach to improve P use efficiency and/or the rate of external Pi acquisition by plants to maximize productivity, growth and/or survival during growth conditions with access to minimal or no P. Such an alternate approach is also urgently required to dramatically reduce the use of P fertilizers, and therefore, to negate the current reliance of modern agriculture on the supply of exogenous P, a rapidly depleting natural resource.

One such molecular approach to provide crops with an enhanced tolerance to cultivation in a P limited environment, or for their growth in other environments where the plant is exposed to other forms of abiotic stress, is the identification of abiotic stress-responsive small non-protein-coding regulatory RNAs (sRNAs) which function at the posttranscriptional level to modulate gene expression [19,20,21,22,23]. In modern molecular research, the application of high throughput sequencing has made profiling of sRNA transcript abundance across plant species, and under altered growth conditions, including exposure of a plant to different abiotic stress regimes, a routine experimental procedure [24,25,26]. For the microRNA (miRNA) class of small regulatory RNA, transcript profiling has identified across multiple evolutionarily diverse plant species a common suite of miRNAs that change in their abundance when essential minerals such as P, copper (Cu), nitrogen (N) and sulfur (S) are lacking from the growth environment [25,26,27]. The responsiveness of a single miRNA to multiple forms of mineral nutrient stress is unsurprising considering the interrelatedness in the complexity of the regulation of mineral nutrient uptake and subsequent transport in plants [24,27,28,29]. In the genetic model plant species *Arabidopsis thaliana* (*Arabidopsis*) for example, the mechanisms of N and P uptake from the soil are reciprocally linked to one another. Therefore, a miRNA with a reduced accumulation profile during N stress, will usually be enhanced in abundance during the cultivation of *Arabidopsis* in a P-depleted environment [20,29,30].

One such miRNA is miR399, with altered miR399 abundance central to the molecular response of *Arabidopsis* to regulate PO_4_ uptake from the soil to maintain P homeostasis [19,20]. More specifically, when the internal supply of P becomes limited in *Arabidopsis* aerial tissues, the transcriptional activity of the five genomic loci (including *MIR399A-MIR399D* and *MIR399E/F*) that encode the precursor transcripts from which the miR399 sRNA is processed, is activated by the PO_4_ stress-responsive transcription factor, PHOSPHATE RESPONSIVE1 (PHR1) [31,32,33]. After precursor transcript processing, the now highly abundant miR399 is transported from the aerial tissues to the roots where it guides the protein machinery of the *Arabidopsis* miRNA pathway to specifically repress the transcript abundance of its target gene, *PHOSPHATE2* (*PHO2*) [19,26,31,34]. PHO2 functions as a ubiquitin conjugating enzyme24 (UBC24), and in *Arabidopsis* roots, PHO2 targets the PO_4_ transporter proteins, PHOSPHATE TRANSPORTER1; 4 (PHT1;4), PHT1; 8 and PHT1; 9 for ubiquitin-mediated degradation [24,35]. The removal of PHT1; 4, PHT1; 8 and PHT1; 9 suppression via the loss of PHO2-mediated regulation, a posttranslational change which itself stems from the enhancement of miR399-directed repression of *PHO2* transcript abundance, promotes root-to-shoot P transport in an attempt by *Arabidopsis* to maintain P homeostasis in its aerial tissues in P limited conditions [3,23,36,37].

In *Arabidopsis*, miR396-directed posttranscriptional regulation of six members of the GROWTH REGULATING FACTOR (GRF) family of transcription factors has been extensively documented to play a central role in all aspects of both the vegetative and reproductive phases of development [38,39,40,41,42,43]. More specifically, miR396-directed regulation of *GRF1*, *GRF2*, *GRF3*, *GRF7*, *GRF8* and *GRF9* gene expression is required for root [38,39], leaf [40,41] and floral organ growth and development [42,43], as well as to control the rate of cell proliferation and to mediate cell aging in *Arabidopsis* [40,41]. In addition to occupying its central role in the control of *Arabidopsis* development, the miR396/*GRF* expression module has also been associated with the adaptive response of *Arabidopsis* and of maize (*Zea mays*), rice (*Oryza sativa*) and wheat (*Triticum aestivum*) to the abiotic stressors, abscisic acid (ABA), cold, drought, heat, osmotic, salt, and ultraviolet light [44,45,46,47,48,49,50]. Alteration of the miR396/*GRF* expression module at the posttranscriptional level across this evolutionary diverse group of plant species, and to such an extensive range of environmental stressors, strongly suggests that this miRNA expression module plays an equally important role in the molecular response of plants to abiotic stress as it does in standard plant development.

Although the miR399/*PHO2* expression module has been repeatedly demonstrated to direct a central role in modulating soil PO_4_ uptake to maintain P homeostasis [9,19,20,31,35], this miRNA/target gene expression module has only been implicated in a small number of other aspects of *Arabidopsis* growth and development. Namely, *Arabidopsis* plants molecularly modified to over-accumulate miR399, and as a direct consequence of this molecular alteration, to have highly reduced *PHO2* expression levels, were also shown to have increased stomata numbers due to elevated stomatal density [51]. In addition, the miR399/*PHO2* expression module has been associated with the early flowering phenotype expressed by *Arabidopsis* plants exposed to reduced ambient temperature during the seedling stage of development [52]. More recently, we demonstrated the requirement of alteration to the abundance of the miR399 sRNA, and of its *PHO2* target transcript, as part of the molecular response of *Arabidopsis* seedlings exposed to salt stress [53]. In direct contrast to the miR399/*PHO2* expression module and the response of *Arabidopsis* to PO_4_ stress, miR396-directed alterations to the abundance of its *GRF* target gene transcripts has been widely documented across an evolutionary diverse group of plant species and environmental stressors. For example, alteration of the abundance of the miR396 sRNA and/or the expression of its *GRF* target genes has been reported in *Arabidopsis*, rice, tobacco (*Nicotiana tabacum*), tomato (*Solanum lycopersicum*), cotton (*Gossypium hirsutum* L.) and creeping bentgrass (*Agrostis stolonifera*) following the exposure of these plant species to salt stress [54,55,56,57,58].

The *Arabidopsis* transformants, including the *MIM396*, *MIR396*, *MIM399* and *MIR399* plant lines, were generated to confirm our initial finding [50] that molecular alteration of the miR396/*GRF* and miR399/*PHO2* expression modules forms part of the molecular response of *Arabidopsis* to salt stress [53,58]. In this study, we exposed 8-day-old *MIM396*, *MIR396*, *MIM399* and *MIR399* seedlings to a 7-day growth period in the absence of PO_4_ to (1) establish the requirement of the miR396/*GRF* expression module as part of the molecular response of *Arabidopsis* to PO_4_ starvation, and (2) confirm the central requirement of the miR399/*PHO2* expression module for *Arabidopsis* to mount an adaptive response to this form of abiotic stress. This approach clearly demonstrated that during the seedling stage of *Arabidopsis* development, unmodified Col-0 plants were the most sensitive to cultivation in a growth environment that lacked P. Accordingly, the experimental approach used here also revealed that both sets of transformant lines which harbored molecular modifications to either the miR396 or miR399 expression module were less sensitive to the imposed stress than were Col-0 seedlings. More specifically, molecular alteration of the miR396 expression module afforded *MIM396* and *MIR396* seedlings a mild degree of tolerance to PO_4_ starvation, whereas the alteration of miR399 abundance via a molecular approach provided *MIM399* and *MIR399* seedlings with a further enhancement in tolerance to this form of abiotic stress. Subsequent quantification of *GRF* and *PHO2* target gene expression strongly suggested that the tolerance displayed by the miR396 and miR399 transformant lines to PO_4_ starvation was the result of altered miRNA target gene expression in *MIM396*, *MIR396*, *MIM399* and *MIR399* seedlings. This study therefore forms an important early step in the future development of molecularly modified plant lines tolerant to cultivation in a growth environment deficient in the level of P required for normal plant growth and development.

## 2. Results

### 2.1. Phenotypic and Physiological Analysis of Control-Grown and Phosphate-Starved Col-0, MIM396, MIR396, MIM399 and MIR399 Seedlings

We have previously demonstrated that the abundance of miRNAs, miR396 and miR399, is significantly altered in 15-day-old *Arabidopsis* whole seedlings following their cultivation for 7 days on *Arabidopsis* growth medium supplemented with 150 millimolar sodium chloride (150 mM NaCl) [50]. In addition, the miR399/*PHO2* expression module has been repeatedly demonstrated to occupy a central role in the molecular response of *Arabidopsis* to PO_4_ starvation [9,19,20,31,35]. We therefore used a molecular modification approach to generate *Arabidopsis* transformant lines with significantly altered miR396 and miR399 abundance. Following experimental verification that miR396 and miR399 abundance was reduced in *MIM396* and *MIR399* plants, respectively, and that the abundance of these two miRNAs was elevated in the *MIR396* and *MIR399* transformant lines [53,58], 8-day-old *MIM396*, *MIR396*, *MIM399* and *MIR399* seedlings, together with unmodified Col-0 seedlings of the same age, were exposed to a 7-day period of PO_4_ starvation. This approach was undertaken to attempt to establish a role for the miR396/*GRF* expression module in the molecular response of *Arabidopsis* to PO_4_ starvation, as well as to compare any established requirement of the miR396/*GRF* expression module to this form of abiotic stress to that of the well documented and central role occupied by the miR399/*PHO2* expression module.

As previously described [53,58], when cultivated for the entire 15-day experimental period on standard *Arabidopsis* growth medium (referred to as P^+^ samples from herein), the growth of *MIM396*/P^+^ and *MIM399*/P^+^ seedlings was promoted (Figure 1C,E), while the developmental progression of the *MIR396*/P^+^ and *MIR399*/P^+^ transformant lines was mildly repressed (Figure 1G,I), compared to Col-0/P^+^ whole seedlings (Figure 1A). Figure 1B shows that the cultivation of 8-day-old Col-0 seedlings for a 7-day period on PO_4_ deficient *Arabidopsis* growth medium (referred to as P^-^ samples from herein) was detrimental to the development of Col-0/P^-^ seedlings. More specifically, the elongation of Col-0/P^-^ rosette leaf petioles was repressed, and the distal tips of Col-0/P^-^ rosette leaves curled down towards the growth medium. In addition, the healthy light-green coloration of Col-0/P^+^ rosette leaves (Figure 1A), was replaced by a dark green to brown coloration in Col-0/P^-^ rosette leaves (Figure 1B). Comparison of Figure 1D to Figure 1C, and Figure 1F to Figure 1E, respectively, shows that the development of the *MIM396* and *MIR396* transformant lines was not impacted to the same degree as it was in 15-day-old Col-0 seedlings by the 7-day cultivation period in the absence of PO_4_. However, the dark green to brown colored pigment still accumulated to a higher level in *MIM396*/P^-^ and *MIR396*/P^-^ rosette leaves, than it did in the rosette leaves of *MIM396*/P^+^ and *MIR396*/P^+^ plants (Figure 1C–F). The developmental progression of the *MIM399* transformant line also appeared to be impeded to a lesser degree by the applied stress (Figure 1G,H), than was the vegetative development of 15-day-old Col-0 seedlings (Figure 1A,B). Namely, the petioles of *MIM399*/P^-^ rosette leaves expanded to the same length as those of *MIM399*/P^+^ rosette leaves, and the area of *MIM399*/P^-^ rosette leaf blades was only slightly reduced compared to those of *MIM399*/P^+^ seedlings. However, the rosette leaves of all *MIM399* seedlings exposed to the 7-day stress treatment period uniformly accumulated high levels of the dark green to brown colored pigment, versus the Col-0 line, where this pigment was observed to only accumulate in the aerial tissues of 50–60% of exposed seedlings (Figure 1B). Of the five *Arabidopsis* lines assessed in this study, the development of the *MIR399* transformant line appeared to be impacted the least by the applied stress, with rosette leaf petiole length, and rosette leaf blade area, as well as the color of *MIR399*/P^-^ rosette leaves, all highly similar to the corresponding phenotypic metrics displayed by *MIR399*/P^+^ plants (Figure 1I,J).

To determine the degree to which altered miR396 or miR399 abundance influenced the phenotypic or physiological response of the *MIM396*, *MIR396*, *MIM399* or *MIR399* transformants to PO_4_ starvation, the phenotypic metrics of fresh weight, rosette area, and primary root length, along with the physiological parameters of anthocyanin abundance and chlorophyll *a* and *b* content were quantitatively assessed (Figure 2). The fresh weight of a Col-0/P^+^ seedling was determined to be 28.1 mg (Figure 2A). As indicated by the developmental phenotypes displayed by 15-day-old control-grown *MIM396* and *MIM399* seedlings (Figure 1C,G), and when compared to Col-0/P^+^ seedlings, the fresh weight of these two transformants was significantly and mildly increased by 20.2% (33.8 mg) and 6.6% (30.0 mg), respectively (Figure 2A). Similarly, via comparison to Col-0/P^+^ seedlings, and as suggested by the developmental phenotypes displayed by *MIR396*/P^+^ and *MIR399*/P^+^ seedlings (Figure 1E,I), the fresh weight of a 15-day-old control-grown *MIR396* and *MIR399* seedling was determined to be mildly and significantly reduced by 4.9% (26.8 mg) and 14.2% (24.1 mg), respectively (Figure 2A). The average fresh weight metric was next used to determine the degree of response of each *Arabidopsis* line to the imposed stress. Compared to the Col-0/P^+^ sample, the average fresh weight of a Col-0/P^-^ seedling was significantly reduced by 32.1% to 19.1 mg (Figure 1B and Figure 2A). Although not reduced to the same degree as determined for Col-0 seedlings, exposure of *MIM396* and *MIR396* seedlings to a 7-day PO_4_ starvation period significantly reduced the fresh weights of *MIM396*/P^-^ (Figure 1D) and *MIR396*/P^-^ (Figure 1F) seedlings by 21.9% (26.4 mg) and 25.0% (20.1 mg), respectively (Figure 2A). Figure 2A also clearly shows that alteration of miR399 abundance, be it either the enhancement or repression of the level of miR399, modulated the degree of response of the *MIM399* and *MIR399* transformants to the applied stress. More specifically, the fresh weight of a *MIM399*/P^-^ or *MIR399*/P^-^ seedling was moderately, yet significantly reduced by 11.6% (26.5 mg) and 10.4% (21.4 mg), respectively, compared to the fresh weight of a *MIM399*/P^+^ and *MIR399*/P^+^ seedling.

The rosette area of 15-day-old control-grown or PO_4_-starved Col-0, *MIM396*, *MIR396*, *MIM399* and *MIR399* whole seedlings was next assessed (Figure 2B) with this analysis revealing that the rosette area of a Col-0/P^-^ seedling was 10.6 mm^2^, which represented a 41.6% reduction in the rosette area of a Col-0/P^+^ seedling at 18.2 mm^2^. Although the rosette area of a *MIM396*/P^+^ seedling was significantly increased by 21.3% to 22.1 mm^2^, compared to a Col-0/P^+^ seedling (Figure 2B), the 7-day stress period was shown to reduce the rosette area of a *MIM396*/P^-^ seedling (14.8 mm^2^) by a similar degree (39.9%) as was determined for the Col-0 sample. The imposed stress induced a milder response in the miR396 and miR399 overexpression lines than it did in either Col-0 or *MIM396* seedlings. More specifically, the rosette area of *MIR396*/P^-^ and *MIR399*/P^-^ seedlings was moderately, yet significantly reduced by 18.8% (13.6 mm^2^) and 17.4% (11.3 mm^2^), from 17.0 and 14.5 mm^2^ for *MIR396*/P^+^ and *MIR399*/P^+^ seedlings, respectively (Figure 1E,I and Figure 2B). Of the five *Arabidopsis* lines analyzed in this study, Figure 2B clearly shows that the lowest degree of negative impact on this phenotypic parameter was observed for the *MIM399* transformant line. More specifically, when compared to Col-0/P^+^ seedlings, the rosette area of a control-grown *MIM399* seedling was moderately increased by 12.6% to 20.5 mm^2^ (Figure 1A,G), however, the rosette area of a *MIM399*/P^-^ seedling was revealed to remain largely unchanged compared to that of a *MIM399*/P^+^ seedling, down by 4.2% to 19.7 mm^2^.

Considering that altered root architecture is a well-documented consequence of the cultivation of *Arabidopsis* in the absence of an adequate supply of P [5,19,59], the primary root length of PO_4_-starved Col-0, *MIM396*, *MIR396*, *MIM399* and *MIR399* seedlings was determined for comparison to the control-grown counterpart of each plant line (Figure 2C). At 47.7 and 48.8 mm, the primary root lengths of *MIR396*/P^+^ and *MIM399*/P^+^ seedlings were highly similar to the primary root length of a 15-day-old control-grown Col-0 seedling at 48.6 mm. This analysis further revealed that compared to Col-0/P^+^ seedlings, the primary root length of a *MIM396*/P^+^ seedling was moderately increased by 9.3% to 53.1 mm, and that the length of the primary root of *MIR399*/P^+^ seedlings was significantly decreased by 12.4% to 42.6 mm (Figure 2C).

The 7-day PO_4_ starvation period had a severe negative impact on primary root elongation of Col-0 seedlings with the length of the primary root of a Col-0/P^-^ seedling significantly reduced by 47.5% to 25.5 mm. Of the four transformant lines analyzed in this study, primary root development was revealed to only be impacted to a similar degree in the miR396 overexpression line, with the length of the primary root of a *MIR396*/P^-^ seedling (24.6 mm) reduced by 47.4% compared to the primary root length of a *MIR396*/P^+^ seedling (Figure 2C). In comparison, primary root elongation was impacted to a much lesser degree in the *MIM396* and *MIR399* transformant lines by the imposed stress. More specifically, the primary root lengths of *MIM396*/P^-^ and *MIR399*/P^-^ seedlings were reduced by 12.7% and 13.5%, respectively, compared to the primary root lengths of their control-grown counterparts, *MIM396*/P^+^ and *MIR399*/P^+^ seedlings (Figure 2C). In contrast to the decreased elongation of the primary root of PO_4_-starved Col-0, *MIM396*, *MIR396* and *MIR399* seedlings, the imposed stress had little to no effect on *MIM399* primary root development with only a very mild 2.5% reduction in primary root length documented for *MIM399*/P^-^ seedlings (Figure 2C), compared to *MIM399*/P^+^ seedlings.

Visual analysis of the aerial tissue phenotypes presented in Figure 1 clearly shows that the most readily apparent variation to rosette development of PO_4_-starved Col-0, *MIM396*, *MIR396*, *MIM399* and *MIR399* seedlings (Figure 1; right hand side panels), compared to the corresponding control-grown counterpart of each *Arabidopsis* line (Figure 1; left hand side panels), was the degree to which a dark green to brown colored pigment accumulated in rosette leaves and rosette leaf petioles of PO_4_-starved seedlings. This prominent darkening in pigmentation was suspected to result from the elevated accumulation of the well-documented abiotic stress associated pigment, anthocyanin [5,60,61,62]. Therefore, spectrophotometry was employed to quantify the abundance of anthocyanin in each *Arabidopsis* line. In the rosette leaves, rosette leaf petioles, and the shoot apex of Col-0/P^+^ seedlings, anthocyanin was determined to accumulate to 2.2 micrograms per gram of fresh weight (µg/g FW). In 15-day-old control-grown *MIM396*, *MIR396*, *MIM399* and *MIR399* seedlings, anthocyanin accumulated to a level equivalent to its abundance in the aerial tissues of Col-0/P^+^ seedlings (Figure 2D). Spectrophotometry next revealed that in the aerial tissues of PO_4_-starved Col-0 seedlings, anthocyanin accumulation was promoted by 60.3% to 3.5 µg/g FW. Anthocyanin production was also promoted, albeit to quite different degrees, in the *MIM396*/P^-^, *MIR396*/P^-^, *MIR399*/P^-^ and *MIR399*/P^-^ samples (Figure 2D). Namely, compared to the control-grown counterpart of each transformant line, the abundance of anthocyanin in the aerial tissues of PO_4_-starved *MIM396*, *MIR396*, *MIR399* and *MIR399* seedlings was promoted by 57.6%, 94.8%, 126.2% and 33.7%, respectively.

As the photosynthetic capabilities of a plant have been demonstrated to be inhibited by exposure to abiotic stress [63], spectrophotometry was also applied to quantify the abundance of the two primary photosynthetic pigments, chlorophyll *a* (Figure 2E) and chlorophyll *b* (Figure 2F). The chlorophyll *a* content of Col-0/P^+^ seedlings was determined to be 0.73 milligrams per gram of fresh weight (mg/g FW). Similarly, and when compared to a Col-0/P^+^ seedling, the chlorophyll *a* content of a control-grown *MIR396* seedling was only mildly reduced by 4.4% to 0.70 mg/g FW. In contrast, the chlorophyll *a* content of *MIM396*/P^+^ and *MIM399*/P^+^ seedlings was moderately elevated by 10.0% and 8.5%, respectively, and the chlorophyll *a* content of the *MIR399*/P^+^ sample was significantly increased by 12.4% (Figure 2E), when compared to the Col-0/P^+^ sample. The 7-day exposure period to PO_4_ starvation was revealed to only mildly alter the chlorophyll *a* content of Col-0/P^-^ (down by 4.4% to 0.70 mg/g FW), *MIR396*/P^-^ (down by 3.5% to 0.67 mg/g FW) and *MIR399*/P^-^ (down by 3.9% to 0.79 mg/g FW) seedlings. In addition, the chlorophyll *a* content of *MIM399*/P^-^ seedlings, when compared to *MIM399*/P^+^ seedlings, was significantly reduced by 10.7% to 0.71 mg/g FW (Figure 2E). In contrast to the reduced abundance of chlorophyll *a* in the PO_4_-starved Col-0, *MIR396*, *MIM399* and *MIR399* samples, the already elevated chlorophyll *a* content of *MIM396*/P^+^ seedlings (0.80 mg/g FW), was further mildly elevated by 3.3% to 0.83 mg/FW in *MIM396*/P^-^ seedlings following the stress treatment period (Figure 2E).

Chlorophyll *b* was determined to accumulate to a much lower level, 0.18 mg/g FW, than chlorophyll *a* (0.73 mg/g FW), in the aerial tissues of 15-day-old Col-0/P^+^ seedlings. In comparison to the value determined for control-grown Col-0 seedlings, spectrophotometry next revealed that the abundance of chlorophyll *b* was significantly elevated by 12.8%, 13.8% and 16.0% in the aerial tissues of the *MIM396*/P^+^, *MIM399*/P^+^ and *MIR399*/P^+^ transformant lines, respectively (Figure 2F). In contrast, the chlorophyll *b* content of *MIR396*/P^+^ seedlings was moderately reduced by 7.6% to 0.17 mg/g FW. Comparison of Col-0/P^-^ seedlings, to Col-0/P^+^ seedlings, showed that the content of chlorophyll *b* was only mildly increased by 5.1% to 0.19 mg/g FW by the applied stress. The content of chlorophyll *b* was elevated by a similar degree in PO_4_-starved *MIR396* seedlings, up by 6.7%, to return the abundance of chlorophyll *b* in the *MIR396*/P^-^ sample to a level equivalent to that originally determined for the Col-0/P^+^ sample at 0.18 mg/g FW (Figure 2F). In contrast to the Col-0/P^-^ and *MIR396*/P^-^ samples, the chlorophyll *b* content remained largely unchanged in the *MIM396*/P^-^ sample, compared to the *MIM396*/P^+^ sample. Furthermore, the content of this primary photosynthetic pigment was revealed by spectrophotometry to be moderately reduced by 9.4% in *MIM399*/P^-^ seedlings, and mildly decreased by 5.8% in *MIR399*/P^-^ seedlings, when compared to its abundance in the *MIM399*/P^-^ and *MIR399*/P^+^ samples, respectively (Figure 2F).

### 2.2. Molecular Profiling of the miR396/GRF Expression Module in Control-Grown and Phosphate-Starved Col-0, MIM396 and MIR396 Seedlings

In *Arabidopsis*, it is well established that the expression of six members of the plant specific *GRF* transcription factor gene family, including *GRF1*, *GRF2*, *GRF3*, *GRF7*, *GRF8* and *GRF9*, are regulated at the posttranscriptional level by miR396 [38,39,40,41,42,43]. Furthermore, each of these miR396 target genes has been documented in *Arabidopsis*, and in other plant species, to play a role in all aspects of vegetative and reproductive development, or in the adaptive response of a plant to environmental stress [38,39,40,41,42,43,44,45,46,47,48,49,50]. Therefore, the RT-qPCR approach was employed to profile miR396 abundance and *GRF* target gene expression to attempt to uncover any change at the molecular level which could have potentially contributed to the unique phenotypic and/or physiological response of *MIM396* and *MIR396* seedlings to PO_4_ starvation. RT-qPCR revealed that compared to Col-0/P^+^ seedlings, miR396 accumulation was significantly reduced and elevated by 3.6- and 2.1-fold in *MIM396*/P^+^ and *MIR396*/P^+^ seedlings, respectively (Figure 3A). Furthermore, and in comparison to the control-grown counterpart of each plant line, PO_4_ starvation significantly decreased the level of the miR396 sRNA 2.6-fold in Col-0/P^-^ and *MIR396*/P^-^ seedlings, and induced a much more moderate decrease (1.1-fold) to the already significantly reduced abundance of the miR396 sRNA in *MIM396*/P^-^ seedlings (Figure 3A).

In control-grown *MIM396*/P^+^ seedlings, reduced miR396 abundance was next revealed by RT-qPCR to result in the expression of the *GRF1*, *GRF2*, *GRF3*, *GRF8* and *GRF9* target genes being significantly elevated 4.7-, 2.8-, 4.2-, 7.8- and 2.3-fold, respectively (Figure 3B–D,F,G). In direct contrast, *GRF7* expression was revealed to be mildly reduced by 1.3-fold in *MIM396*/P^+^ seedlings, compared to its level of expression in Col-0/P^+^ seedlings (Figure 3E). However, when considered together, RT-qPCR readily revealed that miRNA-directed mRNA cleavage was the predominant mode of RNA silencing directed by the miR396 sRNA in 15-day-old *Arabidopsis* whole seedlings to regulate the expression of its *GRF* target genes. In control-grown *MIR396* seedlings, RT-qPCR revealed that elevated miR396 abundance (up by 2.1-fold) mildly decreased the expression level of *GRF1*, *GRF2* and *GRF3* by 1.3-, 1.2- and 1.4-fold, respectively (Figure 3B–D). In addition, the expression of *GRF8* was highly reduced by 2.0-fold in response to the 2.1-fold elevated abundance of miR396 in *MIR396*/P^+^ seedlings (Figure 3A,F). In contrast to the reduced expression trend documented for the *GRF1*, *GRF2*, *GRF3* and *GRF8* target genes, *GRF7* expression remained unchanged (Figure 3E), and the level of *GRF9* expression was mildly elevated by 1.4-fold (Figure 3G) in *MIR396*/P^+^ seedlings, compared to the expression level of these two miR396 target genes in the Col-0/P^+^ sample.

The applied stress significantly decreased the accumulation level of miR396 by 2.6-fold in Col-0/P^-^ seedlings, compared to its level in Col-0/P^+^ seedlings (Figure 3A). It was therefore surprising that reduced miR396 abundance in Col-0/P^-^ seedlings resulted in a significant decrease in the level of expression of all six of its *GRF* target genes. More specifically, *GRF1*, *GRF2*, *GRF3*, *GRF7*, *GRF8* and *GRF9* transcript abundance was significantly reduced in Col-0/P^-^ seedlings by 2.0-, 2.4-, 2.6-, 2.8-, 5.6- and 2.4-fold, respectively (Figure 3B–G), compared to the abundance of each transcript in Col-0/P^+^ seedlings. This unexpected target gene expression trend indicated that the transcriptional activity of all components of the miR396/*GRF* expression module were negatively influenced in 15-day-old *Arabidopsis* whole seedlings by the 7-day PO_4_ starvation treatment period.

When compared to the *MIM396*/P^+^ sample, the mild 1.1-fold decrease in miR396 abundance in *MIM396*/P^-^ seedlings was revealed by RT-qPCR to (1) mildly reduce *GRF1* expression by 1.1-fold (Figure 3B), (2) moderately reduce *GRF2*, *GRF7* and *GRF9* expression by 1.5-, 1.5- and 1.6-fold (Figure 3C,E,G), and (3) significantly decrease the abundance of the *GRF3* and *GRF8* transcripts by 2.0- and 3.8-fold (Figure 3D,F), respectively. As suggested by the expression trends noted in PO_4_-starved Col-0 seedlings, the global decrease in the abundance of all assessed components of the miR396/*GRF* expression module in PO_4_-starved *MIM396* seedlings (Figure 3A–G) again suggested that the transcriptional activity of the entire miR396/*GRF* expression module is negatively impacted by this form of abiotic stress.

RT-qPCR next revealed that the significant decrease in miR396 abundance in *MIR396*/P^-^ seedlings (down by 2.6-fold), compared to the *MIR396*/P^+^ sample (Figure 3A), resulted in a mild reduction in the level of expression of the *GRF1* (1.3-fold), *GRF2* (1.1-fold), *GRF3* (1.2-fold) and *GRF7* (1.3-fold) target transcripts (Figure 3B–E), and extended the expression repression (down by 2.1-fold) to *GRF9* (Figure 3G). Only the *GRF8* target gene showed an expected expression trend in the *MIR396*/P^-^ sample with the abundance of the *GRF8* target transcript moderately increased by 1.7-fold, when compared to the level of expression of *GRF8* in the *MIR396*/P^+^ sample (Figure 3G). However, when taken together, the molecular profiling of the miR396/*GRF* expression module in Col-0, *MIM396* and *MIR396* plants repeatedly showed that the transcriptional activity of all components of this miRNA expression module were repressed by the 7-day PO_4_ starvation period.

### 2.3. Molecular Profiling of the miR399/PHO2 Expression Module in Control-Grown and Phosphate-Starved Col-0, MIM396, MIR396, MIM399 and MIR399 Seedlings

As part of the molecular response of *Arabidopsis* to growth in a PO_4_-depleted environment, the expression of the locus which encodes transcription factor PHR1, has been demonstrated to be mildly promoted [28,31,32,33]. The level of expression of this PO_4_ stress-responsive transcription factor was therefore quantified by RT-qPCR in control-grown and PO_4_-starved Col-0, *MIM396*, *MIR396*, *MIM399* and *MIR399* seedlings. This expression analysis was performed to determine at which point in the PO_4_ starvation response pathway the miR396/*GRF* and miR399/*PHO2* expression modules were potentially imposing their effects. Figure 4A shows that compared to the Col-0/P^+^ sample, *PHR1* expression was moderately increased by 1.5-fold in both the miR396 and miR399 eTM (endogenous target mimic) transformant lines, and to be significantly elevated by 2.0-fold in the miR396 overexpression line following the cultivation of these four *Arabidopsis* lines for the entire 15-day experimental period under standard growth conditions. In contrast to elevated *PHR1* expression in *MIM396*/P^+^, *MIR396*/P^+^ and *MIM399*/P^+^ seedlings, the expression of *PHR1* was mildly repressed by 1.3-fold in the *MIR399*/P^+^ sample (Figure 4A). RT-qPCR next revealed that following PO_4_ starvation, *PHR1* expression was mildly elevated by 1.3-fold in Col-0/P^-^ seedlings. In the four assessed transformant lines, an opposing expression trend was observed. More specifically, *PHR1* transcript abundance was reduced by 1.4-, 2.1-, 5.5- and 1.1-fold in *MIM396*/P^-^, *MIR396*/P^-^, *MIM399*/P^-^ and *MIR399*/P^-^ seedlings, respectively (Figure 4A).

The highly conserved plant miRNA, miR399, is a well-documented PO_4_-responsive miRNA that modulates the expression of *PHO2* in response to an inadequate cellular concentration of P, with previous studies demonstrating that the miR399/*PHO2* expression module plays an essential role in the uptake of Pi from the soil, as well as being required for the maintenance of P homeostasis in *Arabidopsis* shoot and root tissues: both essential processes needed to ensure that a plant can respond to an environment with either limited or deficient levels of P [9,19,20,26,31,34,35]. The expression of the miR399 precursor encoding locus *MIR399A* (Figure 4B), the abundance of the miR399 sRNA (Figure 4C), and that of its target transcript *PHO2* (Figure 4D), were therefore next quantified by RT-qPCR in the *MIM396*, *MIR396*, *MIM399* and *MIR399* transformant lines for comparison to Col-0 seedlings.

The expression of the miR399 precursor encoding locus *MIR399A* (Figure 4B), the abundance of the miR399 sRNA (Figure 4C), and that of its target transcript *PHO2* (Figure 4D), were therefore next quantified by RT-qPCR in the *MIM396*, *MIR396*, *MIM399* and *MIR399* transformant lines for comparison to Col-0 seedlings. When compared to the level of *PRE-MIR399A* expression in the Col-0/P^+^ sample, RT-qPCR revealed that the abundance of the miR399 precursor transcript, *PRE-MIR399A*, was significantly reduced by 2.1-fold in the *MIM396*/P^+^ sample, and remained unchanged in *MIR396*/P^+^ seedlings (Figure 4B). Figure 4B also clearly shows that the transcriptional activity of the *MIR399A* locus was highly induced by the 7-day PO_4_ starvation period, with the level of the *PRE-MIR399A* precursor transcript significantly elevated by 20.9-, 37.0- and 69.4-fold in Col-0/P^-^, *MIM396*/P^-^ and *MIR396*/P^-^ seedlings, respectively, when compared to the level of expression of this miR399 precursor transcript in Col-0/P^+^, *MIM396*/P^+^ and *MIR396*/P^+^ seedlings. RT-qPCR was next applied to quantify miR399 sRNA abundance in control-grown and PO_4_-starved Col-0, *MIM399* and *MIR399* seedlings (Figure 4C). When compared to the Col-0/P^+^ sample, miR399 abundance was shown to be significantly reduced by 2.2-fold in the *MIM399*/P^+^ sample, and to be significantly elevated by 3.6-fold in *MIR399*/P^+^ seedlings, respectively (Figure 4C). In PO_4_ starved seedlings, miR399 sRNA abundance was significantly elevated by an equivalent level, up by 2.1-fold, in the Col-0/P^-^ and *MIM399*/P^-^ samples, and to be induced by a more moderate, yet significant degree (up by 1.4-fold) in the *MIR399*/P^-^ sample (Figure 4C).

Having assessed miR399 precursor transcript or sRNA abundance via RT-qPCR profiling, we next sought to determine what influence a change in the abundance of these transcripts had on the expression of the miR399 target gene, *PHO2* (Figure 4D). This analysis showed that *PHO2* expression was significantly elevated by 3.0- and 3.7-fold in control-grown *MIM396* and *MIR396* seedlings, respectively, compared to its abundance in Col-0/P^+^ seedlings (Figure 4D). In contrast to the *PHO2* expression trend constructed for the two miR396 transformant lines, *PHO2* abundance was revealed by RT-qPCR to be reduced by 2.4-fold in the *MIM399*/P^+^ sample, and to remain largely unchanged in the *MIR399*/P^+^ sample. The 7-day cultivation period on *Arabidopsis* growth medium which lacked PO_4_, repressed the degree of *PHO2* gene expression in all five of the *Arabidopsis* lines assessed in this study. More specifically, RT-qPCR revealed that *PHO2* expression was reduced by 6.7-, 1.4-, 2.3-, 13.7- and 2.8-fold in PO_4_-starved Col-0, *MIM396*, *MIR396*, *MIM399* and *MIR399* seedlings, respectively (Figure 4D).

Post documentation of the global reduction in *PHO2* expression in PO_4_-starved Col-0, *MIM396*, *MIR396*, *MIM399* and *MIR399* seedlings, RT-qPCR was next employed to determine if reduced *PHO2* transcript levels, and therefore reduced PHO2 protein abundance, in turn induced any change in the level of expression of the three loci known to encode the downstream targets of PHO2-mediated ubiquitination, specifically the PO_4_ transporter proteins PHT1;4, PHT1;8 and PHT1;9 [24,26,31,35,36,37]. Compared to its expression level in the Col-0/P^+^ sample, *PHT1;4* transcript abundance was elevated by 4.1-, 3.5- and 1.7-fold in *MIM396*/P^+^, *MIR396*/P^+^ and *MIM399*/P^+^ seedlings, respectively. However, in the *MIR399*/P^+^ sample, *PHT1;4* expression was mildly repressed by 1.2-fold (Figure 4E). When compared to the control-grown counterpart of each assessed line, *PHT1;4* expression was revealed to be significantly elevated by 14.9-, 13.8-, 22.3-, 10.5- and 13.1-fold in PO_4_-starved Col-0, *MIM396*, *MIR396*, *MIM399* and *MIR399* seedlings, respectively (Figure 4E). The PO_4_ transporter *PHT1;8* formed the second target of PHO2-mediated ubiquitination assessed at the posttranscriptional level via the RT-qPCR approach in 15-day-old control and PO_4_-starved Col-0, *MIM396*, *MIR396*, *MIM399* and *MIR399* seedlings. Compared to Col-0/P^+^ seedlings, *PHT1;8* expression was mildly reduced in *MIM396*/P^+^, *MIR396*/P^+^ and *MIR399*/P^+^ seedlings by 1.6-, 2.1- and 1.4-fold, respectively, and in the *MIM399*/P^+^ sample, *PHT1;8* transcript abundance remained at wild-type equivalent levels (Figure 4F). In striking contrast to the mild alterations to *PHT1;8* expression observed in control-grown plants, the abundance of the *PHT1;8* transcript was significantly elevated by 49.3-, 309.1-, 548.2-, 38.2- and 18.4-fold in PO_4_-starved Col-0, *MIM396*, *MIR396*, *MIM399* and *MIR399* seedlings, respectively (Figure 4F). RT-qPCR profiling next revealed that the transcript abundance of the PO_4_ transporter, *PHT1;9*, was mildly elevated in *MIM396*/P^+^ (1.2-fold), *MIR396*/P^+^ (1.2-fold) and *MIM399*/P^+^ (1.4-fold) seedlings, and to only be decreased in its abundance level (down by 1.4-fold) in the *MIR399*/P^+^ sample, when compared to the level of *PHT1;9* expression observed in the Col-0/P^+^ sample (Figure 4G). As demonstrated by RT-qPCR for the *PHT1;4* and *PHT1;8* transcripts in 15-day-old Col-0, *MIM396*, *MIR396*, *MIM399* and *MIR399* seedlings post stress application, the abundance of the *PHT1;9* transcript was elevated in all five assessed *Arabidopsis* lines. More specifically, *PHT1;9* expression was determined to be significantly increased by 4.6-, 8.3- and 12.9-fold in Col-0/P^-^, *MIM396*/P^-^ and *MIR396*/P^-^ seedlings, and to be moderately elevated by 2.3- and 3.0-fold in the *MIM396*/P^-^ and *MIR399*/P^-^ samples, respectively (Figure 4G).

## 3. Discussion

Due to the major annual losses of up to 40% in total yield of the world’s most important crop species stemming from their cultivation on soils which lack an appropriate supply of P [10,11,12,13,14,15], an alternate approach is required to address limited P availability impacting global crop production to attempt to match agricultural outputs with the expanding consumer demands of an increasing world population [16,17,18]. One such approach is the generation of new crop lines that harbor molecular modifications that provide the plant with a heightened tolerance to abiotic stress. Towards this goal, here we report on the consequence of molecularly altering the abundance of the highly conserved miRNAs, miR396 and miR399, in the genetic model plant species, *Arabidopsis*. We have previously demonstrated that the abundance of these two miRNAs is significantly altered in *Arabidopsis* following the exposure of *Arabidopsis* seedlings to salt stress [50,53,58]. Therefore, considering the well-established requirement of the miR399/*PHO2* expression module in the adaptive response of *Arabidopsis* to its cultivation in PO_4_-limited or depleted conditions [9,19,20,26,31,34,35], together with the knowledge that the miR396/*GRF* expression module forms part of the molecular response of *Arabidopsis*, and of other plant species to a vast array of abiotic stressors [38,39,40,41,42,43,44,45,46,47,48,49,50], here we exposed 8-day-old Col-0, *MIM396*, *MIR396*, *MIM399* and *MIR399* seedlings to a 7-day PO_4_ starvation period. Specifically, these two sets of transformant lines engineered to have altered miR396 and miR399 abundance, respectively, were exposed to this stress regime in parallel to each other to (1) attempt to establish the requirement of the miR396/*GRF* expression module in the molecular response of *Arabidopsis* to PO_4_ starvation, and to (2) compare the miR396 findings to those obtained for *Arabidopsis* transformant lines which harbored molecular alterations to the expression module of the well characterized PO_4_ stress responsive miRNA, miR399.

It was interesting to observe that when cultivated under a standard *Arabidopsis* growth regime for the entire experimental period, both of the eTM transformant lines displayed enhanced vegetative development (Figure 1A,C,G). More specifically, compared to 15-day-old Col-0/P^+^ seedlings, the fresh weight and rosette area of *MIM396*/P^+^ seedlings were significantly enhanced by 20.2% and 21.3%, respectively. Similarly, the fresh weight and rosette area of control-grown *MIM399* seedlings were increased by 6.6% and 12.6%, respectively (Figure 2A,B). Enhanced vegetative development of the miR396-specific eTM transformant line most likely stemmed from reduced miR396 abundance, and therefore, deregulated *GRF* target gene expression, with Figure 3A showing that miR396 abundance was significantly reduced in *MIM396*/P^+^ seedlings compared to Col-0/P^+^ seedlings, and that the expression of its *GRF1*, *GRF2*, *GRF3* and *GRF9* target genes were significantly elevated (Figure 3B–D). Previous research has assigned roles in leaf growth and development in *Arabidopsis*, specifically the control of leaf morphology, to the GRF1, GRF2, GRF3 and GRF9 transcription factors [40,64,65,66]. Namely, GRF1, GRF2 and GRF3 have been shown to promote leaf development [40,64,65], while the GRF9 transcription factor acts as a negative regulator of leaf development [66]. Therefore, the effects of elevated *GRF9* transcript abundance in control-grown *MIM396* seedlings appeared to have been negated by the greater collective enhancement to the level of expression of the *GRF1*, *GRF2* and *GRF3* target genes, which in turn, promoted the vegetative development of 15-day-old *MIM396*/P^+^ seedlings. RT-qPCR also revealed that compared to the Col-0/P^+^ sample, the abundance of the miR399 sRNA and its *PHO2* target transcript were reduced by 2.2- and 2.4-fold, (Figure 4C,D), and that the expression of the PO_4_ transporters, *PHT1;4* and *PHT1;9* were mildly elevated by 1.7- and 1.4-fold, respectively (Figure 4E,G), in the miR399-specific eTM transformant line. When these Figure 4 expression profiles are considered together with those results previously reported by others [35,37,67,68,69], the mild promotion of *MIM399* vegetative development likely stemmed from reduced PHO2-directed ubiquitin-mediated repression of PHT1 transporter activity, thereby leading to the enhancement of Pi root-to-shoot translocation, where the now-abundant P was unloaded into the juvenile rosette leaves and utilized as an additional cellular resource for vegetative growth.

Compared to the developmental progression of 15-day-old control-grown Col-0 seedlings (Figure 1A), and in contrast to the two eTM transformant lines assessed in this study (Figure 1C,G), the development of the miR396 and miR399 overexpression lines when cultivated under a standard *Arabidopsis* growth regime was mildly inhibited (Figure 1E,I). The fresh weight and rosette area reductions recorded for the *MIM396*/P^+^ sample (Figure 2A,B), were most likely a consequence of enhanced miR396-directed repression of *GRF1*, *GRF2* and *GRF3* gene expression. More specifically, an elevated miR396 level in the *MIR396*/P^+^ sample resulted in mild reductions to the abundance of the *GRF1*, *GRF2* and *GRF3* transcripts (Figure 3A–D). In *Arabidopsis*, these three GRF transcription factors have been shown to act as positive regulators of leaf development [40,64,65,66]. Therefore, the RT-qPCR documented 1.3-, 1.2- and 1.4-fold reduction in the expression level of *GRF1*, *GRF2* and *GRF3*, in response to the 2.1-fold elevation in miR396 abundance could, in part, account for the mildly reduced progression of the vegetative development of *MIR396*/P^+^ seedlings (Figure 1E). Further, our demonstration that the expression of the negative leaf development regulator, *GRF9*, was mildly elevated by 1.4-fold in control-grown *MIR396* seedlings in response to an increased abundance of the miR396 sRNA, adds further weight to the proposal that the specific molecular modifications made to the miR396/*GRF* expression module as part of this study, modifications which were demonstrated to alter *GRF* target gene expression, were likely the direct cause of the developmental trajectory change observed for *MIM396*/P^+^ seedlings.

The repressed developmental progression of *MIR399*/P^+^ seedlings, compared to Col-0/P^+^ seedling development (Figure 1A), is clearly shown in Figure 1I, and further confirmed via quantification of the phenotypic metrics, fresh weight (Figure 2A) and rosette area (Figure 2B). Molecular profiling of miR399 abundance, the expression of the miR399 target gene, *PHO2*, and the expression of the targets of PHO2-directed ubiquitination, *PHT1;4*, *PHT1;8* and *PHT1;9*, readily revealed the cause of the repression of the vegetative development of *MIR399* seedlings. Namely, in response to the 3.6-fold elevation of miR399 accumulation (Figure 4C), *PHO2* transcript abundance was mildly repressed by 1.1-fold (Figure 4D), as was the expression of the PO_4_ transporters, *PHT1;4* (down 1.2-fold), *PHT1;8* (down 1.4-fold) and *PHT1;9* (down 1.4-fold) (Figure 4E–G). PHT1 PO_4_ transporter proteins are required for Pi acquisition from the soil and the subsequent translocation of Pi from the roots to the shoot tissues via the xylem [37,67,69]. Further, once the translocated Pi is unloaded from the xylem into photosynthetically active leaves, the P is either directly utilized by the source tissue or, alternatively, is loaded into the phloem for distribution to other non-photosynthetically active shoot organs such as juvenile leaves and the reproductive tissues [67,68,69]. Therefore, reduced *PHT1* gene expression, which would result in repressed PHT1 PO_4_ transporter activity in the miR399 overexpression line, would likely reduce the availability of P in the aerial tissues of this transformant line, leading to the observed repression to the vegetative development of *MIR399*/P^+^ seedlings.

Comparison of the phenotypes displayed by 15-day-old control-grown and PO_4_-starved Col-0, *MIM396*, *MIR396*, *MIM399* and *MIR399* seedlings (Figure 1), when taken together with the quantified phenotypic metrics of fresh weight (Figure 2A), rosette area (Figure 2B), and primary root length (Figure 2C), clearly revealed Col-0 seedlings to be the most sensitive to the applied stress, and the *MIM399* transformant line to be the least sensitive to cultivation for a 7-day period on *Arabidopsis* growth medium which lacked PO_4_. More specifically, the fresh weight, rosette area and primary root length of Col-0/P^-^ seedlings were reduced by 32.1%, 41.6% and 47.5%, respectively, when compared to the corresponding phenotypic metrics of Col-0/P^+^ seedlings (Figure 2A–C). In contrast, the fresh weight, rosette area and primary root length of *MIM399*/P^-^ seedlings only showed mild reductions of 11.6%, 4.2% and 2.5%, respectively, compared to the corresponding phenotypic metrics of the *MIM399*/P^+^ sample (Figure 2A–C). The expression of the PO_4_ stress responsive transcription factor, *PHR1* [31,32,33], was revealed by RT-qPCR to be mildly elevated by 1.2-fold in Col-0/P^-^ seedlings, and to be significantly reduced by 5.5-fold in *MIM399*/P^-^ seedlings, compared to its level of expression in the control-grown counterpart of these two *Arabidopsis* lines (Figure 4A). Interestingly, although *PHR1* was revealed to have an opposing expression trend in Col-0 and *MIM399* seedlings post application of the stress treatment regime, miR399 abundance was elevated by the same degree, up by 2.1-fold, in these two *Arabidopsis* lines. This finding strongly suggests that the PHR1 transcription factor exerts little to no regulatory influence on the transcriptional activity of the miR399 encoding loci, *MIR399A* to *MIR399F*, in *Arabidopsis* seedlings at 15-days of age in response to PO_4_ starvation. Although miR399 abundance was increased in Col-0/P^-^ and *MIM399*/P^-^ seedlings by the same degree, it is important to note that comparison of the relative expression level of the miR399 sRNA in these two PO_4_-starved plant lines revealed that the level of miR399 was at a 2.4-fold lower abundance in the *MIM399*/P^-^ sample (RE = 0.94 ± 0.11) than in the Col-0/P^-^ sample (RE = 2.11 ± 0.03) (Figure 4C). Accordingly, *PHO2* expression was found to be decreased by 6.7- and 13.7-fold in the Col-0/P^-^ and *MIM399*/P^-^ samples, respectively (Figure 4D), in response to elevated miR399 abundance (Figure 4C). Again, however, direct comparison of the level of relative expression of *PHO2* further revealed that the abundance of the *PHO2* transcript was 5.0-fold lower in the *MIM399*/P^-^ sample (RE = 0.03 ± 0.01) than it was in the Col-0/P^-^ sample (RE = 0.15 ± 0.03). Direct comparison of miR399 and *PHO2* transcript abundance in PO_4_-starved Col-0 and *MIM399* plants (Figure 4C,D), implied that the degree of release of PHO2-mediated posttranslational regulation of the activity of the PHT1 PO_4_ transporter proteins would be far greater in *MIM399*/P^-^ seedlings than in the Col-0/P^-^ sample.

The level of expression of the three loci that encode the PO_4_ transporter targets of PHO2-directed ubiquitination, namely *PHT1;4*, *PHT1;8* and *PHT1;9*, was therefore subsequently assessed via RT-qPCR and revealed that the expression was higher for all three PO_4_ transporters in Col-0/P^-^ seedlings than it was in *MIM399*/P^-^ seedlings (Figure 4E–G), even though *PHO2* expression was demonstrated to be lower in PO_4_-starved *MIM399* seedlings than in Col-0/P^-^ seedlings (Figure 4D). If enhanced P translocation from the roots to the aerial tissues to be used as an additional cellular resource was providing the miR399-specific eTM transformant line with the observed degree of tolerance to growth in a PO_4_ deplete environment, the RT-qPCR analyses conducted in this study failed to uncover such a direct causative link (Figure 4). Considering that PHT1;4, PHT1;8 and PHT1;9 form targets of PHO2-mediated ubiquitination at the posttranslational level [37,67,68,69], the Figure 4 findings indicate that additional analyses at the protein level are required to definitively identify the molecular mechanism that rendered Col-0 plants the most sensitive to the imposed stress, and the *MIM399* transformant line the most tolerant to growth in a PO_4_-deficient environment. Alternatively, a recent study by Shukla et al. [70] demonstrated that the exogenous application of mild doses of P (i.e., 1.25 to 10.00 mM) promoted *Arabidopsis* seedlings to develop rosettes of significantly larger size. Therefore, a future avenue of research would be to repeat these analyses [70] on Col-0 and *MIM399* seedlings to determine the degree to which the P response pathway is defective in the *MIM399* transformant line that harbors a molecularly modified miR399/*PHO2* expression module.

Further evidence which suggested that the promoted vegetative development of the *MIM399* transformant line was masking the actual degree to which the P response pathway was defective in the miR399-specific eTM plant line was the demonstration that the well-characterized stress pigment, anthocyanin [5,60,61,62], accumulated to a much higher level in 15-day-old PO_4_-starved *MIM399* seedlings than it did in either Col-0/P^-^, *MIM396*/P^-^, *MIR396*/P^-^ or *MIR399*/P^-^ seedlings (Figure 1). More specifically, comparison of the quantified abundance of anthocyanin in PO_4_-starved Col-0, *MIM396*, *MIR396*, *MIM399* and *MIR399* seedlings, to the control-grown counterpart of each *Arabidopsis* line (Figure 2D), revealed anthocyanin accumulation to be promoted by 60.3%, 57.6%, 94.8%, 126.2% and 33.7%, respectively. This finding directly opposes the phenotypic data presented in Figure 2A–C to suggest that the molecular modifications made to the miR399/*PHO2* expression module in the *MIM399* transformant line; that is, eTM transgene-directed repression of miR399 accumulation had indeed altered the ability of *MIM399* seedlings to appropriately respond to growth in a PO_4_-deficient environment. This quantitative analysis also revealed that the accumulation of anthocyanin was promoted to the lowest degree in *MIR399* seedlings post application of the 7-day stress treatment regime. In *Arabidopsis*, P stress induces the gibberellin-DELLA pathway which in turn promotes the anthocyanin biosynthesis pathway [71]. Therefore, the contrast of the level to which anthocyanin accumulated in *MIM399*/P^-^ and *MIR399*/P^-^ seedlings tentatively suggests that any molecular modification introduced into *Arabidopsis* which alters miR399 abundance, and therefore *PHO2* target gene expression, interferes with the ability of *Arabidopsis* to appropriately respond to growth in a PO_4_-deficient environment.

The physiological assessments performed here also revealed another interesting result: the abundance of the two primary photosynthetic pigments, chlorophyll *a* and *b*, were significantly elevated by 12.4% and 16.0% in *MIR399*/P^+^ seedlings, respectively (Figure 2E,F), compared to their abundance in Col-0/P^+^ seedlings. An elevated chlorophyll content in 15-day-old *MIR399*/P^+^ seedlings formed a highly curious result considering that we [53] and others [19,23,35] have previously demonstrated that the overexpression of miR399, or conversely, the efficient repression of *PHO2* expression (i.e., the *pho2* mutant), is highly detrimental to the later stages of vegetative development of the *Arabidopsis* lines that harbor such modifications. Perturbed vegetative development in these lines is the result of the over-accumulation of P in the aerial tissues, leading to an overall reduction in rosette area and the development of areas of chlorosis and necrosis in mature rosette leaves [19,23,35,53]. Although chlorophyll *a* and *b* content were elevated to their highest levels in *MIR399*/P^+^ seedlings, when all five assessed *Arabidopsis* lines were cultivated under a standard *Arabidopsis* regime for the entire 15-day experimental period, the vegetative development of this transformant line was also repressed to the greatest extent (Figure 1 and Figure 2A–C). The molecular profiling of PO_4_ transporter expression in *MIM396*/P^+^, *MIR396*/P^+^, *MIM399*/P^+^ and *MIR399*/P^+^ seedlings for comparison to control-grown Col-0 seedlings (Figure 4E–G), revealed that *PHT1;4* and *PHT1;9* expression was only reduced in the *MIR399*/P^+^ sample (Figure 4E,G). Considering that the PHT1;4 and PHT1;9 PO_4_ transporters play a central role in P acquisition from the soil, and its subsequent translocation from the roots to the shoots via the xylem [37,67,68,69], reduced *PHT1;4* and *PHT1;9* expression would potentially limit the amount of P available for routine molecular functions to continue normally in the cells of *MIR399*/P^+^ rosette leaves. Therefore, the observed enhancement to the abundance of chlorophyll *a* and *b* in *MIM399*/P^+^ seedlings may be the result of a physiological response to combat any adverse effects stemming from a possible reduction in the content of available P in *MIM399*/P^+^ rosette leaves.

One of the most striking findings stemming from this study was the global repression of all assessed components of the miR396/*GRF* expression module following the cultivation of 8-day-old Col-0 seedlings for a 7-day period on *Arabidopsis* growth medium which lacked PO_4_ (Figure 3). More specifically, compared to Col-0/P^+^ seedlings, the abundance of miR396, and the expression of its six *GRF* target genes, including *GRF1*, *GRF2*, *GRF3*, *GRF7*, *GRF8* and *GRF9*, were all demonstrated to be significantly reduced in Col-0/P^-^ seedlings (Figure 3). As stated above, the GRF1, GRF2 and GRF3 transcription factors all play a positive regulatory role in *Arabidopsis* leaf development [40,64,65,66]. Therefore, reduced *GRF1*, *GRF2* and *GRF3* expression in Col-0/P^-^ seedlings (Figure 3B–D) formed an expected result considering the severe impact to the vegetative development of 15-day-old Col-0 seedlings following the application of the 7-day stress treatment regime (Figure 1 and Figure 2). Furthermore, reduced *GRF1*, *GRF2* and *GRF3* expression, would most likely have also masked any positive influence on rosette leave development stemming from decreased *GRF9* transcript abundance (Figure 3G), with GRF9 previously assigned a negative regulatory role in *Arabidopsis* leaf development [66]. Decreased *GRF7* expression in Col-0/P^-^ seedlings also formed an expected result, with previous research performed on the *Arabidopsis grf7* single mutant associating defective GRF7 transcription factor activity with a heighten tolerance to a range of abiotic stresses [46]. The authors proposed that the abiotic stress tolerance displayed by the *grf7* single mutant was likely to be the result of the elevated expression of a large and functionally diverse cohort of stress-responsive genes [46]. In addition to PO_4_-starved Col-0 seedlings, *GRF7* transcript abundance was also reduced in the *MIM396*/P^-^ and *MIR396*/P^-^ samples; a finding that further supports the requirement of repressed *GRF7* gene expression as part of the molecular response of *Arabidopsis* to P stress, as has been shown previously for other abiotic stressors such as drought and salt stress [46]. The expression of the sixth miR396 target gene profiled in this study, *GRF8*, was also reduced in Col-0/P^-^ seedlings (Figure 3F). However, to date, a functional role in either *Arabidopsis* development, or the response of *Arabidopsis* to other forms of abiotic or biotic stress, has not been assigned to the GRF8 transcription factor. It is important to note here, that of the six miR396 target genes profiled in Col-0/P^-^ seedlings, the expression of *GRF8* was repressed by the greatest degree, a 5.6-fold reduction in transcript abundance (Figure 3F). This finding strongly implies that repressed *GRF8* gene expression, in parallel to reduced *GRF1*, *GRF2*, *GRF3*, *GRF7* and *GRF9* transcript abundance, forms an important part of the molecular response of *Arabidopsis* to PO_4_ starvation during the seedling stage of vegetative development. It is important to note here that we have previously profiled all components of the miR396/*GRF* expression module in 15-day-old Col-0 seedlings post their 7-day exposure to two other forms of abiotic stress, specifically cadmium (Cd) and salt stress [58,72]. Exposure of 15-day-old Col-0 seedlings to either elevated levels of Cd or NaCl, revealed unique expression changes for each component of the miR396 expression module, including decreased, unchanged, or elevated miR396 accumulation or *GRF* target gene expression [58,72]. Transcript-specific expression alterations under conditions of elevated Cd or NaCl [58,72], versus the uniform decrease in the transcriptional activity of all components of the miR396/*GRF* expression module post PO_4_ starvation (Figure 3), strongly implies the absolute requirement of repressed expression module transcriptional activity for *Arabidopsis* to mount a molecular, and potentially adaptive response, to PO_4_ starvation.

## 4. Materials and Methods

### 4.1. Plant Expression Vector Construction and Transformant Line Generation

The construction of the plant expression vectors used to transform wild-type *Arabidopsis* plants (ecotype Columbia-0 (Col-0)) for the generation of the four transformant lines analyzed in this study has been described in detail previously [53,58]. In brief, the plant expression vectors, p*AtMIM396*, p*AtMIR396*, p*AtMIM399* and p*AtMIR399*, were produced via the placement of artificially synthesized (Integrated DNA Technologies, Sydeny, Australia) DNA fragments behind the *Cauliflower mosaic virus* (CaMV) 35S promoter of the pBART plant expression vector. More specifically, for the construction of the p*AtMIM396* target mimicry transgene, the endogenous non-cleavable miR399 target site of the *Arabidopsis* non-protein-coding RNA, *INDUCED BY PHOSPHATE STARVATION1* (*IPS1*; *AT3G09922*), was replaced with a miR396a-specific non-cleavable target site according to the design method outlined in [73,74]. For the miR399-specific eTM transgene, the *IPS1* sequence was artificially synthesized without any requirement for further design modification. To generate the miR396 and miR399 overexpression transgenes, p*AtMIR396* and p*AtMIR399* respectively, DNA fragments were artificially synthesized to exactly match the non-protein-coding RNA sequences of the *Arabidopsis* miRNA precursor transcripts, *PRE-MIR396A* (*AT2G10606*) and *PRE-MIR399C* (*AT5G62162*).

Post their generation, the resulting p*AtMIM396*, p*AtMIR396*, p*AtMIM399* and p*AtMIR399* plant expression vectors were subsequently introduced into *Agrobacterium tumefaciens* (*Agrobacterium*) strain GV3101 and these *Agrobacterium* cultures were used to transform wild-type *Arabidopsis* plants according to the protocol of [75]. In the T_2_ generation of transformants, the (1) transgene copy number, and the (2) zygosity of each transformant was determined via a standard PCR-based genotyping approach. Of the T_2_ transformants determined to be homozygous for a single chromosome insertion event, the ‘best performing’ transformant line was selected to represent the *MIM396*, *MIR396*, *MIM399* and *MIR399* transformant populations for subsequent phenotypic, physiological and molecular analyses, with all experimentation reported here conducted on the T_3_ transformant generation. It is important to note here that the best performing *MIM396*, *MIR396*, *MIM399* and *MIR399* transformant line was selected via RT-qPCR analysis of miRNA abundance in the T_2_ generation, that is; identification of the (1) *MIM396* and *MIM399* transformant lines with the greatest degree of reduced miR396 and miR399 abundance, and (2) *MIR396* and *MIR399* transformant lines with the highest degree of elevated miR396 and miR399 abundance.

### 4.2. Plant Material and Plant Growth

The seeds sourced from Col-0, *MIM396*, *MIR396*, *MIM399* and *MIR399* plants were surface sterilized via incubation in a sealed chamber for 90 min (min) at room temperature with chlorine gas. The sterilized seeds were plated out onto standard *Arabidopsis* growth medium (half-strength Murashige and Skoog (MS) salts), and post sealing each plate with gas permeable tape, the plates were transferred to 4 °C and incubated in the dark for 48 h for stratification. Post stratification, the plates were transferred to a temperature-controlled growth cabinet (A1000 Growth Chamber, Conviron^®^, Melbourne, Australia) and cultivated for an 8-day period under a standard *Arabidopsis* growth regime of 16 h light/8 h dark, and a day/night temperature of 22/18 °C. At 8 days of age, an equal number (*n* = 48) of Col-0, *MIM396*, *MIR396*, *MIM399* and *MIR399* seedlings were transferred to either (1) fresh standard *Arabidopsis* growth medium (P^+^ plants; control treatment), or (2) *Arabidopsis* growth medium where the PO_4_ had been replaced with an equivalent molar amount (1.0 millimolar (mM)) of potassium chloride (KCl) (P^-^ plants; PO4 starvation treatment). Post seedling transfer, the control and PO_4_ starvation plates were sealed with gas permeable tape and returned to the temperature-controlled growth cabinet. The control and PO_4_ starved Col-0, *MIM396*, *MIR396*, *MIM399* and *MIR399* seedlings were then cultivated for an additional 7-day period under a standard *Arabidopsis* growth regime.

### 4.3. Phenotypic and Physiological Assessment of 15-Day-Old Control-Grown and PO_4_-Starved Col-0, MIM396, MIR396, MIM399 and MIR399 Seedlings

To allow for the direct comparison of the *MIM396*, *MIR396*, *MIM399* and *MIR399* transformant lines to 15-day-old Col-0 seedlings, each performed phenotypic and physiological assessment was converted to a percentage. Taking this approach, control-grown Col-0 seedlings were assigned a value of 100% for each assessed metric. Rosette area (millimeters squared (mm^2^)) was calculated via the analysis of photographic images of control-grown and PO_4_-starved Col-0, *MIM396*, *MIR396*, *MIM399* and *MIR399* seedlings grown on media plates which were orientated horizontally for the entire 15-day experimental period with the freely available software, ImageJ. Similarly, primary root length (millimeters (mm)) was determined via ImageJ analysis of photographic images of control-grown and PO_4_-starved 15-day-old Col-0, *MIM396*, *MIR396*, *MIM399* and *MIR399* seedlings which were grown on vertically orientated plates for the 7-day treatment period post the transfer of 8-day-old seedlings to fresh *Arabidopsis* growth medium. The fresh weight of 15-day-old control-grown or PO_4_-starved Col-0, *MIM396*, *MIR396*, *MIM399* and *MIR399* seedlings was determined via the transfer of 12 pooled seedlings of each assessed plant line into pre-weighed 1.5 mL microfuge tubes. Each sample tube was immediately capped post seedling transfer and then the weight of the sample tube was recalculated. The initial weight was then subtracted from the final weight of each 1.5 mL microfuge tube, and this value was subsequently divided by 12 to give the average fresh weights of 15-day-old control-grown or PO_4_-starved Col-0, *MIM396*, *MIR396*, *MIM399* and *MIR399* seedlings. This procedure was repeated four times for each plant line, and growth condition, and post calculation of the final weight of each tube, the tubes were immediately submerged in liquid nitrogen (LN_2_) followed by storage at −80 °C for future use as the four biological replicates for the RT-qPCR analyses reported here in Figure 3 and Figure 4.

The physiological parameters of anthocyanin abundance and chlorophyll *a* and *b* content were also calculated for the *MIM396*, *MIR396*, *MIM399* and *MIR399* transformant lines for their direct comparison to 15-day-old control-grown Col-0 seedlings. To determine the anthocyanin abundance of each *Arabidopsis* line, 100 mg of freshly harvested rosette leaves were ground into a fine powder in LN_2_. Once the powder had completely thawed, 1.0 milliliter (mL) of acidic methanol (1.0% (*v/v*) of 12 N hydrogen chloride (HCl)) was added to each sample. The samples were briefly vortexed to thoroughly mix the ground plant material into solution, and then incubated at 4 °C for 2 h. Any remaining cellular debris was pelleted out of solution via centrifugation at 15,000 × *g* for 5 min at room temperature. The absorbance (A) at 530 (A_530_) and 657 (A_657_) nanometers (nm) of each sample was then measured in a GENESYS 10S spectrophotometer (ThermoFisher Scientific, Brisbane, Australia) with acidic methanol used as the blanking solution. The abundance of anthocyanin in each sample, in units of milligrams per gram of fresh weight (mg/g FW), was calculated using the equation; A_530_ − 0.25 × A_657_/fresh weight (g), according to [76]. To determine the chlorophyll *a* and *b* content of control-grown and PO_4_-starved Col-0, *MIM396*, *MIR396*, *MIM399* and *MIR399* seedlings, 100 mg of freshly harvested rosette leaves was ground into a fine powder in LN_2_. To this powder, 1.0 mL of 80% (*v/v*) ice-cold acetone was added and each sample was thoroughly mixed by careful hand inversion. The samples were then incubated in the dark for 24 h at room temperature. Following this incubation period, any cellular debris that remained in solution was removed via centrifugation at 15,000 × *g* for 5 min at room temperature. The A at wavelengths 646 (A_646_) and 663 (A_663_) nm was then determined in a GENESYS 10S spectrophotometer with the resuspension agent, 80% (*v/v*) acetone, used as the blanking solution. The chlorophyll *a* and *b* content (micrograms per gram of fresh weight (μg/g FW)) of each sample was then determined using the Lichtenthaler’s equations exactly as outlined in [77].

### 4.4. RT-qPCR Assessment of miRNA Abundance and Target Gene Expression in 15-Day-Old Control-Grown and PO_4_-Starved Col-0, MIM396, MIR396, MIM399 and MIR399 Seedlings

To extract total RNA from control-grown and PO_4_-starved 15-day-old Col-0, *MIM396*, *MIR396*, *MIM399* and *MIR399* seedlings, TRIzol™ Reagent was used according to the manufacturer’s (Invitrogen™, Brisbane, Australia) protocol. A NanoDrop^®^ spectrophotometer (NanoDrop^®^ ND-1000, ThermoFisher Scientific, Brisbane Australia) was used to determine the concentration of each sample and the quality of each total RNA extraction was ensured via the use of a standard electrophoresis approach on a 1.2% (*w/v*) ethidium bromide-stained agarose gel. Four biological replicates consisting of pools of 12, 15-day-old control-grown and PO_4_-starved Col-0, *MIM396*, *MIR396*, *MIM399* and *MIR399* seedlings were subsequently used as templates to synthesize complementary DNA (cDNA). The synthesis of a cDNA product specific to either the miR396 or miR399 sRNA was conducted as previously reported in [50] using a protocol adapted from [78], and a corresponding high molecular weight (HMW) cDNA library was synthesized from the same set of samples as previously described in [79,80]. All RT-qPCR assessments of miRNA abundance or mRNA expression were conducted using the cycling conditions; (1) 1 × 95 °C for 10 min, and (2) 45 × 95 °C for 10 s and 60 °C for 15 s. In addition, the GoTaq^®^ qPCR Master Mix (Promega, Sydney, Australia) was used as the fluorescent reagent for all performed RT-qPCR experiments. Following the RT-qPCR assessments, miRNA abundance or mRNA expression was quantified using the 2^−ΔΔCT^ method. The small nucleolar RNA, *snoR101*, and the housekeeping gene, *UBIQUITIN10* (*UBI10*; *AT4G05320*), were used to normalize miRNA abundance and mRNA expression, respectively, in control-grown or PO_4_-starved *MIM396*, *MIR396*, *MIM399* and *MIR399* transformants, for comparison to the Col-0/P^+^ sample. Appendix A lists the sequence of each DNA oligonucleotide used in the RT-qPCR analyses reported in this study.

### 4.5. Statistical Analysis

As stated above, the phenotypic, physiological, and molecular data reported in this study were obtained from the analysis of four biological replicates with each biological replicate consisting of 12 pooled seedlings. Statistical analysis was performed using the one-way analysis of variance (ANOVA; RRID:SCR_002427) method while the Tukey’s *post hoc* test was performed using the SPSS program (IBM, Armonk, United States; RRID: SCR_002865). The results of the statistical analyses are presented as letters above the columns on the relevant histograms in Figure 2, Figure 3 and Figure 4. The same letter above a histogram column indicates a non-statistically significant difference (*p* > 0.05), whereas a different letter above a histogram column indicates a statistically significant difference (*p* < 0.05).

## 5. Conclusions

Considering that the molecular and metabolic processes of nucleic acid synthesis, phospholipid production, coenzyme activation and the generation of the vast volume of chemical energy required to drive these processes all rely on an adequate supply of the essential macronutrient, phosphorous (P), an alternate approach to develop new plant lines with a decreased reliance upon P is of extreme importance if agriculture is to continue to achieve the outputs required by an ever increasing population under less favorable environmental conditions in the future. One such approach is the use of molecular technologies to alter key steps in nutrient response pathways. At the posttranscriptional level, the highly conserved plant miRNAs, miR396 and miR399, have been repeatedly demonstrated to mediate central roles as part of the molecular response of a diverse range of evolutionarily diverse plant species to an array of environmental stressors. Here we demonstrate that *Arabidopsis* transformant lines molecularly modified to harbor alterations to the miR396 and miR399 expression modules are less sensitive to growth in a PO_4_ deficient environment than were unmodified wild-type *Arabidopsis* plants at the seedling stage of vegetative development. The global reduction in all assessed transcript components of the miR396/*GRF* expression module identified a previously unreported requirement for this miRNA, and potentially, all six of its *GRF* target genes as part of the molecular response of *Arabidopsis* to PO_4_ starvation. We also confirm that *Arabidopsis* transformant lines with a molecularly altered miR399/*PHO2* expression module are more resistant to cultivation in a growth environment lacking PO_4_. More specifically, RT-qPCR suggested that the PO_4_ stress tolerance afforded to the *Arabidopsis* miR399 transformant lines was likely the result of decreased *PHO2* transcript abundance, and therefore, defective PHO2-mediated ubiquitination of the PO_4_ transporters, PHT1;4, PHT1;8 and PHT1;9. In turn, enhanced activity of these three PO_4_ transporters would promote root-to-shoot P translocation, and subsequently; an elevated level of available P in *Arabidopsis* aerial tissues would enable enhanced cellular function. When taken together, the results presented in this study represent an early, yet important step in the use of molecular technologies to develop new *Arabidopsis* lines that are tolerant to growth in an environment where P is either in limited supply or is completely lacking. Furthermore, due to the high level of conservation of the miR396 and miR399 expression modules in plants, future advances made in *Arabidopsis* hold a degree of promise for their translation into agronomically important crop species as part of the development of new plant lines that can be cultivated in an environment that lacks an adequate supply of P.

## Figures and Tables

**Figure 1 plants-10-02570-f001:**
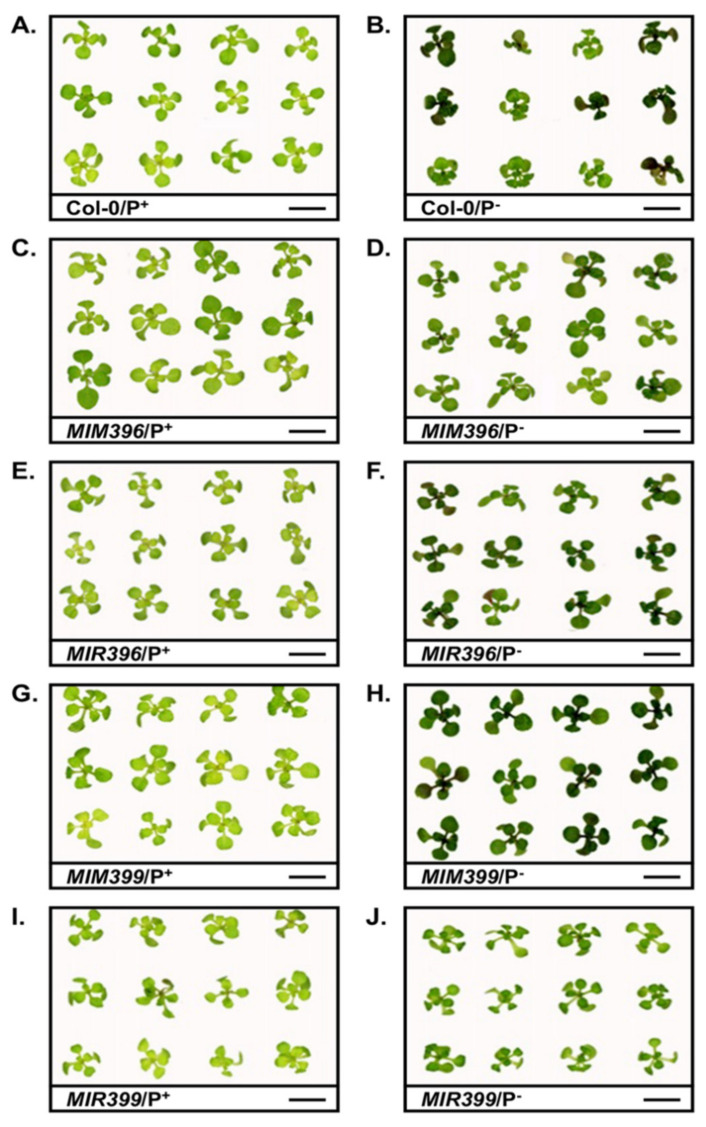
Phenotypes displayed by 15-day-old control-grown and PO_4_-starved Col-0, *MIM396*, *MIR396*, *MIM399* and *MIR399* seedlings. Phenotypes displayed by 15-day-old (**A**) Col-0/P^+^, (**B**) Col-0/P^-^, (**C**) *MIM396*/P^+^, (**D**) *MIM396*/P^-^, (**E**) *MIR396*/P^+^, (**F**) *MIR396*/P^-^, (**G**) *MIM399*/P^+^, (**H**) *MIM399*/P^-^, (**I**) *MIR399*/P^+^, and (**J**) *MIR399*/P^-^ seedlings. (**A**–**J**) Scale bar = 1.0 cm.

**Figure 2 plants-10-02570-f002:**
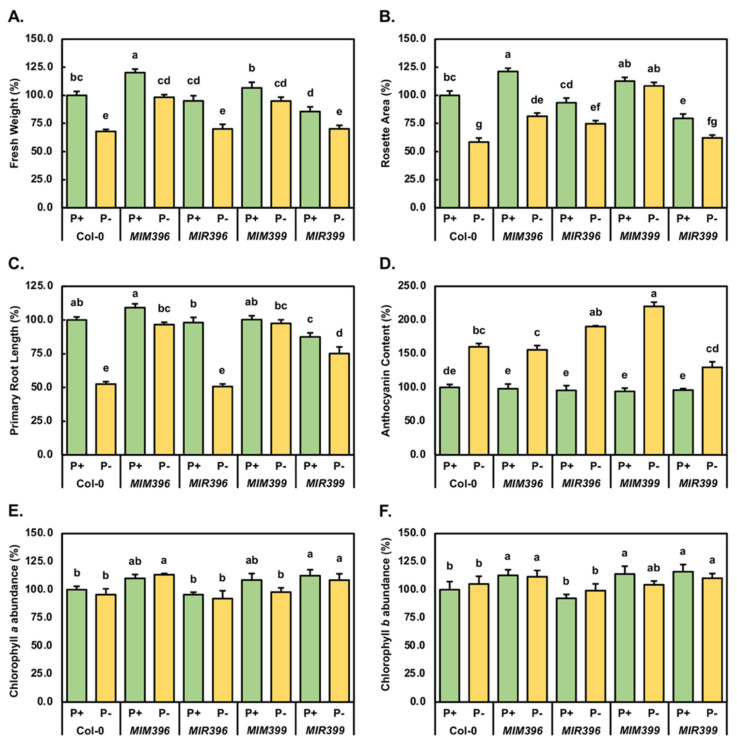
Phenotypic and physiological assessment of 15-day-old control-grown and PO_4_-starved Col-0, *MIM396*, *MIR396*, *MIM399* and *MIR399* seedlings. Quantification of the phenotypic parameters of (**A**) fresh weight (mg), (**B**) rosette area (mm^2^) and (**C**) primary root length (mm) for control-grown and PO_4_-starved 15-day-old Col-0, *MIM396*, *MIR396*, *MIM399* and *MIR399* seedlings. Quantification of the physiological parameters of (**D**) anthocyanin content (µg/g FW), (**E**) chlorophyll *a* and (**F**) chlorophyll *b* abundance (mg/g FW), for control-grown and PO_4_-starved 15-day-old Col-0, *MIM396*, *MIR396*, *MIM399* and *MIR399* seedlings. All phenotypic and physiological metrics are presented as a percentage (%) post comparison to the respective values obtained for Col-0/P^+^ plants which were assigned a value of 100% for each assessed growth characteristic. (**A**–**F**) Error bars represent the standard deviation of four biological replicates. The statistical data were analyzed with the use of the one-way ANOVA and Tukey’s *post hoc* tests and a statistically significant difference (*p*-value < 0.05) is denoted by a different letter above a histogram column.

**Figure 3 plants-10-02570-f003:**
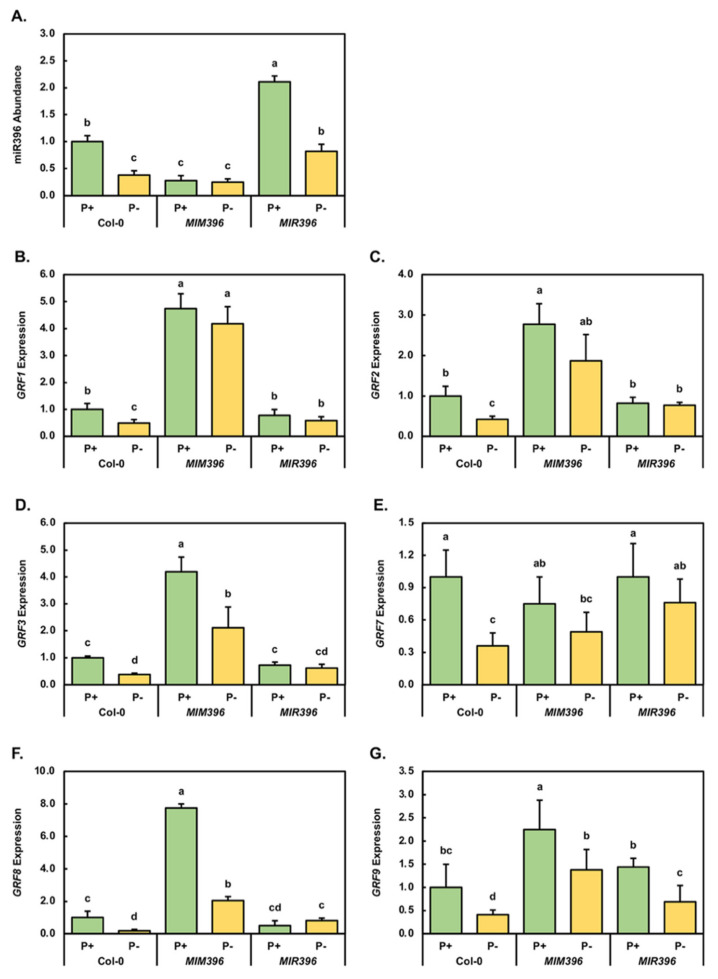
Molecular profiling of the miR396/*GRF* expression module in 15-day-old control-grown and PO_4_-starved Col-0, *MIM396* and *MIR396* seedlings. (**A**) Quantification of miR396 abundance by RT-qPCR in control-grown (P^+^) and PO_4_-starved (P^-^) Col-0, *MIM396* and *MIR396* plants. RT-qPCR quantification of the expression of the six members of the *Arabidopsis GRF* gene family targeted by miR396 for expression regulation, including *GRF1* (**B**), *GRF2* (**C**), *GRF3* (**D**), *GRF7* (**E**), *GRF8* (**F**) and *GRF9* (**G**), in the Col-0/P^+^, Col-0/P^-^, *MIM396*/P^+^, *MIM396*/P^-^, *MIR396*/P^+^ and *MIR396*/P^-^ samples. (**A**–**G**) Error bars represent the standard deviation of four biological replicates. All statistical data were analyzed via the use of the one-way ANOVA and Tukey’s *post hoc* tests. A statistically significant difference (*p*-value < 0.05) is represented by a different letter above a histogram column.

**Figure 4 plants-10-02570-f004:**
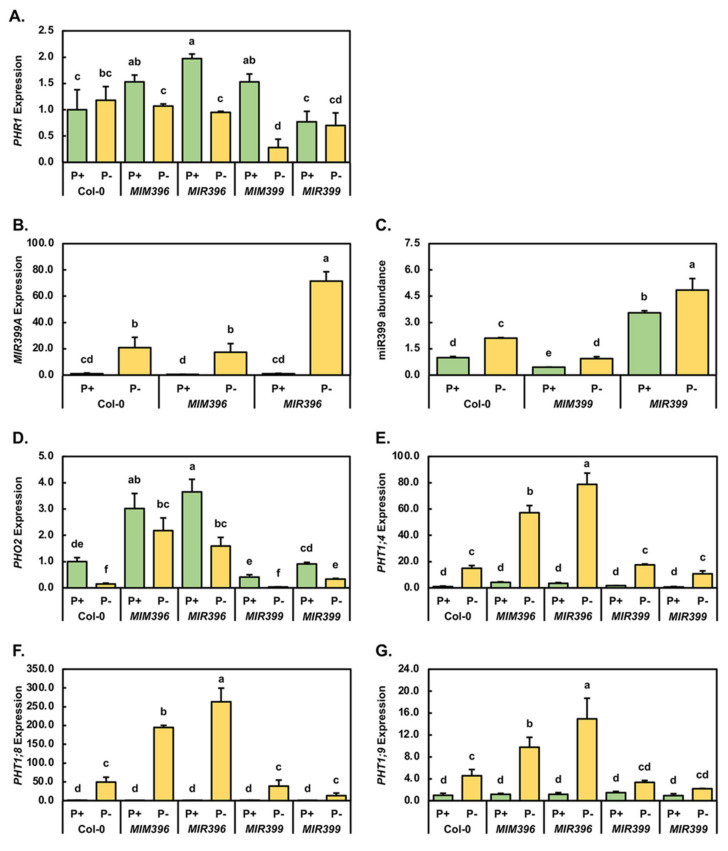
Molecular profiling of upstream and downstream components of the miR399/*PHO2* expression module in 15-day-old control-grown and PO_4_-starved Col-0, *MIM396*, *MIR396*, *MIM399* and *MIR399* seedlings. RT-qPCR quantification of *PHR1* expression (**A**), *MIR399A* gene expression (**B**), miR399 abundance (**C**), *PHO2* expression (**D**) and the level of gene expression of the PO_4_ transporters, *PHT1;4* (**E**), *PHT1;8* (**F**) and *PHT1;9* (**G**) in Col-0/P^+^, Col-0/P^-^, *MIM396*/P^+^, *MIM396*/P^-^, *MIR396*/P^+^, *MIR396*/P^-^, *MIM399*/P^+^, *MIM399*/P^-^, *MIR399*/P^+^ and *MIR399*/P^-^ seedlings. (**A**–**G**) The error bars above histogram columns represent the standard deviation of four biological replicates and all statistical data were analyzed using the one-way ANOVA and Tukey’s *post hoc* tests. A different letter above a histogram column represents a statistically significant difference (*p*-value < 0.05).

## Data Availability

Available upon request to the authors. This includes the seeds from the generated *MIM396*, *MIR396*, *MIM399* and *MIR399* transformant lines.

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
