# Peer review of "Molecular Manipulation of the miR396 and miR399 Expression Modules Alters the Response of Arabidopsis thaliana to Phosphate Stress"

_plants, 2021, doi:10.3390/plants10122570_

Round 1
Reviewer 1 Report
In the manuscript entitled ¨Molecular Manipulation of the miR396 and miR399 Expression Modules Alters the Response of Arabidopsis thaliana to Phosphate Stress¨ By Pegler and collaborators, authors show that altering miR396 and miR399 abundance conferred tolerance to Phosphate starvation, and also show that this tolerance correlates with the altered expression of diverse mRNA targets either of miR396 or miR399.
Minor comments, observations
Overstatements
Although some conclusions of the manuscript are sustained by the presented results, I´ve found conclusions that might be consider as overstatements, such as the following:
¨this study provides the first step in the future development of new plant lines via a molecular modification approach which are tolerant to P stress¨
Major comments
Figures/Figure legends
A graphic in figure 1 explaining the molecular alterations in the Arabidopsis transformants, including the MIM396, MIR396, MIM399 and MIR399 148 plant lines is indispensable, and explain in the text in detail each of the lines related to miRN399 and MiR396, citing reference 50 is not enough. Authors assume that the readers will read in detail the cited papers where they generated MIR396, MIR399, MIM396 and MIM399, this is and independent story and the genomic background related to both miRNAs should be explained in the text and figures of THIS article.
Moreover, they should correlate the modification that occurred in each of the 4 backgrounds with the phenotypes described, for example:
MIM3XX which has a Loss of Gain of function of MiRXX showed XXX phenotype, and shoud correlate with MIRXX phenotype and molecular background. This should be taken into consideration along all the results section, as well as in the Dicussion sections.
Besides quantitations photos of the differences in root lenght among lines ant treatments should be included.
Author Response
Comments and Suggestions for Authors
In the manuscript entitled ¨Molecular Manipulation of the miR396 and miR399 Expression Modules Alters the Response of Arabidopsis thaliana to Phosphate Stress¨ By Pegler and collaborators, authors show that altering miR396 and miR399 abundance conferred tolerance to Phosphate starvation, and also show that this tolerance correlates with the altered expression of diverse mRNA targets either of miR396 or miR399.
Minor comments, observations
Overstatements
Although some conclusions of the manuscript are sustained by the presented results, I´ve found conclusions that might be consider as overstatements, such as the following:
¨this study provides the first step in the future development of new plant lines via a molecular modification approach which are tolerant to P stress¨
** The authors thank the reviewer for identifying this issue. We have thoroughly edited the original submission of our study in order to address this concern in the revised submission.
Major comments
Figures/Figure legends
A graphic in figure 1 explaining the molecular alterations in the Arabidopsis transformants, including the MIM396, MIR396, MIM399 and MIR399 148 plant lines is indispensable, and explain in the text in detail each of the lines related to miRN399 and MiR396, citing reference 50 is not enough. Authors assume that the readers will read in detail the cited papers where they generated MIR396, MIR399, MIM396 and MIM399, this is and independent story and the genomic background related to both miRNAs should be explained in the text and figures of THIS article.
** The authors wish to thank the Reviewer for identifying this oversight. An extensive explanation regarding the construction of the four assessed transformant lines has been added to the Materials and Methods section of the revised manuscript (specifically, please see lines 789 to 821 of page 18/24 of the revised manuscript). We have also added the required references (Franco-Zorrilla JM et al., 2007; Todesco M et al., 2010; Clough and Bent, 1998) to the newly included text and to the Reference List (citations 73, 74 and 75 in the revised manuscript).
**However, the technologies used in this study to generate the miR396 and miR399 knockdown and overexpression lines have been extensively described in the literature by others, and therefore, we do not believe that the addition of a schematic to Figure 1 would be of any benefit to the readership of this study.
Moreover, they should correlate the modification that occurred in each of the 4 backgrounds with the phenotypes described, for example:
MIM3XX which has a Loss of Gain of function of MiRXX showed XXX phenotype, and should correlate with MIRXX phenotype and molecular background. This should be taken into consideration along all the results section, as well as in the Discussion sections.
**Please see the above related comment: the generation and in planta expression of MIM and MIR transgenes for miRNA knockdown and overexpression, respectively, are widely used technologies to manipulate the abundance of a specific miRNA/s, both in the plant and animal systems. Moreover, those that research gene silencing in plants, or work within the greater area of RNAi in other organisms, would be highly familiar with how these two technologies work. The authors therefore fail to see any benefit to the provision of additional text to continually explain to the reader the molecular alterations introduced to each plant line throughout all sections of the manuscript. In fact, provision of such text would likely distract from molecular findings being reported in the manuscript.
Besides quantitations photos of the differences in root length among lines and treatments should be included.
** The primary root length measurements reported in Figure 2C of control and PO4 stressed plants is a quantified assessment. The authors are therefore of the opinion that reporting this data in a quantified, and statistically analysed format, is more appropriate than the presentation of images of plant roots.
Reviewer 2 Report
The authors of the manuscript " Molecular Manipulation of the miR396 and miR399 Expression 2 Modules Alters the Response of Arabidopsis thaliana to Phos-3 phate Stress " presents a very interesting study. The development of new plant lines that will be less sensitive to the lack of available P is one of the important directions of research to ensure food production at a sufficient level in less favorable environmental conditions. The article is written in a clear and understandable way. The selection of literature is appropriate and logical, there are no unnecessary citations.
Author Response
The authors of the manuscript " Molecular Manipulation of the miR396 and miR399 Expression 2 Modules Alters the Response of Arabidopsis thaliana to Phos-3 phate Stress " presents a very interesting study. The development of new plant lines that will be less sensitive to the lack of available P is one of the important directions of research to ensure food production at a sufficient level in less favorable environmental conditions. The article is written in a clear and understandable way. The selection of literature is appropriate and logical, there are no unnecessary citations.
**Dear Reviewer,
The authorship team wish to thank you for your highly positive review of our study.
Your positive feedback is very much appreciated by the authors.
Reviewer 3 Report
A well-done piece of work that is of importance when issues with climate change and stress adaption in plants need to be understood and addressed for bettering crop productivity. Research such as this, adds to the unravelling of the unknowns in the non-coding regions. It would be good to see this research translated to any crop plant and studied until harvest, to check for produce quality. This would give a good standing/completion to understanding at least in part, the yield parameters in phosphorous stressed in plants, because ultimately it is the produce that we are interested in for consumption. As mentioned in the discussion, further studies on ubiquitination (or proteome) and probably some metabolomics could also reveal interesting plant stress strategies.
Kindly check the manuscript for minor slips such as Line 212: ‘appeared to be impacted..’
Figure 1: Kindly label the plates on the outside so that the labeling is clearly visible. Also place the bars so that they don’t overlap the plants.
Author Response
A well-done piece of work that is of importance when issues with climate change and stress adaption in plants need to be understood and addressed for bettering crop productivity. Research such as this, adds to the unravelling of the unknowns in the non-coding regions. It would be good to see this research translated to any crop plant and studied until harvest, to check for produce quality. This would give a good standing/completion to understanding at least in part, the yield parameters in phosphorous stressed in plants, because ultimately it is the produce that we are interested in for consumption. As mentioned in the discussion, further studies on ubiquitination (or proteome) and probably some metabolomics could also reveal interesting plant stress strategies.
Kindly check the manuscript for minor slips such as Line 212: ‘appeared to be impacted..’
Figure 1: Kindly label the plates on the outside so that the labeling is clearly visible. Also place the bars so that they don’t overlap the plants.
** Dear Reviewer,
The authorship team thank you for your positive and insightful review of our study.
** For your interest: co-author Eamens is currently in the initial stages of the process of translating the research findings made in this study relating to the Arabidopsis miR399/PHO2 expression module to Brassica napus (canola), an agronomically important cropping species in Australia. Canola possesses a highly conserved miR399/PHO2 expression module to that of Arabidopsis. It will therefore be of high interest to determined whether similar molecular alteration of canola results in the expression of desirable phenotypes under environmental stress.
** We have thoroughly edited the revised vision of our study to address the grammatical errors that you indicated in your review of our original manuscript.
** We have also made the requested changes to Figure 1, and the authors wish to thank the reviewer for identifying this issue.
** Again, thank you for your positive support of this work.
Reviewer 4 Report
The authors reported studies of "phophate resistance" of established miR396,miR399 relevant transgenic plants. They presented experimental results following to their recent publications. Thus they might think they need less explanation to this work but I felt that the ms needs more explanation to be more reader-friendly and more appealing to general readers.
- Are they using only one line for respective MIM 396, MIR396, MIM399, MIR399 transgenic plants? I do not think they need more cautious interpretation and experimental strategy to draw some conclusions based on transgenic plants. Control plants should be those of transformant line of empty vector, to be strict.
- Figure 2: The authors performed statistical analysis and made some discussions. But I cannot be confident that they conducted adequate analysis. Is one-way ANOVA the right one for them to make discussion in this context? They should compare responses of each line with control plants.
- The authors should provide the explanation of PO4 stress in their studies. And what concentration of PO4 gives rise to the stress? In the Methods, the authors described a simple comment on this point. PO4 had been replaced with an equivalent molar amount (1 mM) of KCl. PO4 is effective even at a range of micromolar concentration. How strict did the authors handle other conponents which might smuggle PO4 into the medium?
- Anthocyanin content might be a too general indicator of stress conditions. The discussion linking the miR function and anthocyanin production might be too tough,
Author Response
The authors reported studies of "phophate resistance" of established miR396,miR399 relevant transgenic plants. They presented experimental results following to their recent publications. Thus they might think they need less explanation to this work but I felt that the ms needs more explanation to be more reader-friendly and more appealing to general readers.
**Dear Reviewer,
Thank you for your above comments. As part of the editing process to generate the revised version of our original submission we have added text to various manuscript sections where we believe more explanation was required
- Are they using only one line for respective MIM 396, MIR396, MIM399, MIR399 transgenic plants? I do not think they need more cautious interpretation and experimental strategy to draw some conclusions based on transgenic plants. Control plants should be those of transformant line of empty vector, to be strict.
- Figure 2: The authors performed statistical analysis and made some discussions. But I cannot be confident that they conducted adequate analysis. Is one-way ANOVA the right one for them to make discussion in this context? They should compare responses of each line with control plants.
- The authors should provide the explanation of PO4stress in their studies. And what concentration of PO4 gives rise to the stress? In the Methods, the authors described a simple comment on this point. PO4 had been replaced with an equivalent molar amount (1 mM) of KCl. PO4 is effective even at a range of micromolar concentration. How strict did the authors handle other conponents which might smuggle PO4 into the medium?
- Anthocyanin content might be a too general indicator of stress conditions. The discussion linking the miR function and anthocyanin production might be too tough,
** Please see our responses to your 4 above comments, listed as A1 through to A4 below;
A1. We thank the Reviewer for identifying this oversight. In the revised version of our manuscript, we have added extensive additional information on transformant line generation and subsequent select of the transformant lines carried through to experimental analysis. We have generated many different Arabidopsis transformant lines as part of the course of our various research endeavours, and as part of these analyses we have not observed any difference in the molecular or phenotypic characteristics of unmodified Arabidopsis versus Arabidopsis plants into which an empty vector control was added. We have therefore ceased this practice in our more recent studies. Furthermore, the developmental phenotypes expressed by the transformant lines characterised in this study display phenotypes similar to those previously reported by other researchers who used a molecular approach to manipulate miR396 or miR399 abundance. We are therefore highly confident that the phenotypes reported here as the direct result of the molecular modifications introduced into Arabidopsis, and are not the result of unwanted or unintended consequences stemming from the introduction of the pBART plant expression vector used.
A2. The authors thank the Reviewer for this suggestion. However, the authors are of the opinion that the 1-way ANOVA used to statistically assess our data in the originally submitted manuscript version is the more appropriate form of statistical analysis of the reported data. Our reasoning: for the non-stressed (P+) samples, the values obtained for the MIM396/P+, MIR396/P+, MIM399/P+ and MIR399/P+ transformant lines are only compared to the corresponding value obtained for the Col-0/P+ sample. Similarly, for the PO4-stressed (P-) samples, the Col-0/P- sample was only compared to the Col-0/P+ sample, the MIM396/P- sample was only compared to the MIM396/P+ sample, the MIR396/P- sample was only compared to the MIR396/P+, the MIM399/P- sample was only compared to the MIM399/P+ sample, and the MIR399/P- sample was only compared to the MIR399/P+. Therefore, on each occasion, only a single variable was assessed. Please also see similar studies where a 1-way ANOVA has been successfully applied: https://www.mdpi.com/1422-0067/20/1/153/htm https://www.mdpi.com/2073-4425/9/10/475/htm https://www.mdpi.com/2223-7747/10/3/452/htm
A3. The authors respectfully disagree with this Reviewer concern as we have already provided clear explanation in the Materials and Methods section of our manuscript on the approach used to generate Arabidopsis growth media which is devoid of PO4 (we are assessing PO4 starvation in this study, not the consequence/s of limited PO4 availability). Please also see similar studies which have used the same approach to that reported here: doi: 10.1186/1471-2229-10-64. // doi: 10.1104/pp.111.175257. // doi: 10.1016/j.plantsci.2015.06.020 // doi: 10.3390/plants8050124. In addition, the authors have extensive experience in the use of solid Arabidopsis growth medium, and of growth medium of other model plant species, and which includes extensive experience in the preparation of all the various components required for the production of these media, to ensure tightly controlled growth conditions as part of our experimental analyses. We are therefore of the opinion that issues relating to growth media preparation played no role in the phenotypic, physiological, or molecular datasets reported in this study. We have also extensively modified the text to indicate that we are assessing PO4 starvation in this study, not reduced PO4 availability. We thank the reviewer for identifying this oversight in our original submission
A4. Again, the authors respectfully disagree with this Reviewer concern. We, and others, have repeatedly, and successfully, used anthocyanin as an appropriate indicator of stress, especially in Arabidopsis plants of numerous genetic backgrounds cultivated on plant growth medium optimised for cultivation of this model species.
Round 2
Reviewer 1 Report
Unfortunatelly the authors failed to improve the manuscript in two major requests.
Rev: Besides quantitations photos of the differences in root length among lines and treatments should be included. Moreover, they should correlate the modification that occurred in each of the 4 backgrounds with the phenotypes described, for example: MIM3XX which has a Loss or Gain of function of MiRXX showed XXX phenotype, and should correlate with MIRXX phenotype and molecular background. This should be taken into consideration along all the results section, as well as in the Discussion sections.
Authors response: **Please see the above related comment: the generation and in planta expression of MIM and MIR transgenes for miRNA knockdown and overexpression, respectively, are widely used technologies to manipulate the abundance of a specific miRNA/s, both in the plant and animal systems. Moreover, those that research gene silencing in plants, or work within the greater area of RNAi in other organisms, would be highly familiar with how these two technologies work. The authors therefore fail to see any benefit to the provision of additional text to continually explain to the reader the molecular alterations introduced to each plant line throughout all sections of the manuscript. In fact, provision of such text would likely distract from molecular findings being reported in the manuscript.
2nd request from reviewer:
Authors should provide clear correlation among the down or up regulation of each miRNA and the phenotypes. Indeed the molecular findings should correlate with the phenotypes, is no a distractor but a necessary complement.
Rev Request: Besides quantitations photos of the differences in root length among lines and treatments should be included.
Author´s response:
** The primary root length measurements reported in Figure 2C of control and PO4 stressed plants is a quantified assessment. The authors are therefore of the opinion that reporting this data in a quantified, and statistically analysed format, is more appropriate than the presentation of images of plant roots.
2nd request from reviewer:
Author´s refuse to present photos of the root lenght phenotypes side by side with the Wt. This is quite dissapointing, specially since this experiment is extremely easy and only takes the days plants reach the age where authors see PRL difference. I insist that this is a crucial experiment and in the case of mutant roots that grow in -P conditions confocal and nomarsky pictures of the root apical meristem are essential to show that roots are not fully differentiated in comparisson with the control.
I hope authors and the Editor can see the relevance of these requests, which are easy to obtain. No major molecular experiment is requested.
Reviewer 4 Report
The authors quite politely responded to the comments raised. I am satisfied with the amendments the authors made in this revision.